# No clear evidence of a difference between individuals who self-report an absence of auditory imagery and typical imagers on auditory imagery tasks

Zoë Pounder[1,2], Alison F. Eardley[1], Catherine Loveday[1], Samuel Evans[1,3] *

1 Department of Psychology, School of Social Sciences, University of Westminster, London, United Kingdom, 2 Department of Experimental Psychology, University of Oxford, Oxford, United Kingdom, 3 Neuroimaging, King's College London, London, United Kingdom

* samuel.d.evans@kcl.ac.uk

**Data Availability Statement:** All files are available from the Open Science Framework (https://osf.io/xgzct/).

## Abstract

Aphantasia is characterised by the inability to create mental images in one's mind. Studies investigating impairments in imagery typically focus on the visual domain. However, it is possible to generate many different forms of imagery including imagined auditory, kinesthetic, tactile, motor, taste and other experiences. Recent studies show that individuals with aphantasia report a lack of imagery in modalities, other than vision, including audition. However, to date, no research has examined whether these reductions in self-reported auditory imagery are associated with decrements in tasks that require auditory imagery. Understanding the extent to which visual and auditory imagery deficits co-occur can help to better characterise the core deficits of aphantasia and provide an alternative perspective on theoretical debates on the extent to which imagery draws on modality-specific or modality-general processes. In the current study, individuals that self-identified as being aphantasic and matched control participants with typical imagery performed two tasks: a musical pitch-based imagery and voice-based categorisation task. The majority of participants with aphantasia self-reported significant deficits in both auditory and visual imagery. However, we did not find a concomitant decrease in performance on tasks which require auditory imagery, either in the full sample or only when considering those participants that reported significant deficits in both domains. These findings are discussed in relation to the mechanisms that might obscure observation of imagery deficits in auditory imagery tasks in people that report reduced auditory imagery.

## Introduction

Mental imagery extends across multiple senses, including auditory, gustatory, olfactory, tactile and motor imagery. Aphantasia is characterised by the absence or reduction of voluntary sensory imagery [1, 2]. While imagery impairments in aphantasia in the visual domain are

**Funding:** The author(s) received no specific funding for this work.

**Competing interests:** The authors have declared that no competing interests exist.

relatively well documented [3–6], non-visual imagery abilities in aphantasia are much less well understood.

Recent studies have asked participants to introspect on their imagery experience across multiple domains and have shown that as well as reporting lower visual imagery, people with aphantasia often report imagery deficits in other, non-visual domains [7–9]. These studies indicate that there is significant heterogeneity in the profile of imagery deficits expressed across modalities between individuals. Indeed, whilst many individuals report imagery deficits across multiple modalities, a small number also report selective deficits in just the visual [9] or auditory domains [10, 11], suggestive of disassociations in imagery ability. This might indicate that there are aphantasia 'sub-types' and has led to a broader discussion of the definition and terminology used to refer to individuals with reduced imagery [1, 11]. At present, we do not know if these self-reported deficits in other domains translate to reductions in behavioural performance on tasks that require imagery in those domains. If this was the case, this would provide stronger evidence for different 'subtypes' of aphantasia, including those with multimodal imagery deficits. More broadly, capturing the extent that imagery deficits manifest in a single or multiple domains in aphantasia may provide an alternative perspective on theoretical debates concerning the extent to which imagery draws on a single unitary or multiple independent cognitive mechanisms [12].

In previous studies, aphantasic individuals have been shown to exhibit reduced priming through visual imagery during binocular rivalry [5], recall fewer objects and episodic memory details during memory recall [3, 13] and reduced physiological responses when imagining frightening stories or bright objects [14, 15]. However, it is not always the case that overt differences in visual imagery have been found when comparing people with aphantasia to typical imagers. This might suggest a disconnect between subjective experience of visual imagery and visual imagery behaviour and more broadly, might align with theoretical accounts suggesting that visual imagery phenomenology does not play a functional role in cognitive processes [16]. Indeed, when individuals with aphantasia have been presented with batteries of tasks thought to involve visual imagery, differences in behaviour have sometimes been minimal. Specifically, differences were apparent only in a subgroup of participants that report the lowest visual imagery, were found in only one or a small number of imagery tasks in the battery, were apparent in response time (but not accuracy) or were shown only at the highest levels of difficulty [4–6, 17, 18]. However, whilst subtle, these differences might equally indicate qualitatively different or weaker imagery, which in turn might be interpreted as evidence against epiphenomenal accounts. These minimal differences are observed despite participants reporting a substantial reduction in their vividness of visual imagery, which one might have expected would lead to differences in performance on tasks that involve imagery.

It is not clear why differences in performance on visual imagery tasks are not more apparent in aphantasic groups; however, there are several possible explanations. For example, people with aphantasia might use compensatory non-visual processes or strategies, such as verbal or spatial codes [3, 4, 6] or exhibit differences in decision making processes [18]. Another may be that they make adequate use of imagery implicitly but lack conscious awareness of their ability [4, 19]. Alternatively, their deficits might be subtler than is suggested by their self-report and the tasks used are not sensitive enough to identify their imagery impairments or else draw on features of imagery that are well preserved in people with aphantasia. The heterogeneity in the profile of imagery abilities in those with aphantasia [9] and in those with typical imagery more broadly [20], raises important questions about the necessity of imagery. Further research, in a broader range of tasks, is required to build a better picture of the capabilities of people with aphantasia, especially in understanding whether self-reported deficits translate to observable differences in behaviour in imagery domains beyond vision.

One non-visual domain in which people with aphantasia often report reduced imagery is audition [7, 10]. The extent to which visual and auditory imagery share the same or different cognitive mechanisms remains unclear. Neuroimaging studies show that auditory and visual imagery activate many of the same brain regions, suggesting that imagery might be supported by a common cognitive mechanism. For instance, the default mode network and pre-frontal cortex are engaged by imagery generation and introspective processes for both auditory and visual imagery [21]. More generally, auditory images are represented in similar ways to visual images. For example, it takes longer to scan through an imagined melody that contains a longer series of beats [22, 23] in a manner analogous to the increased time that it takes to visually scan imagined objects that are further apart in distance [24, 25] or rotate objects that require larger angular rotations [26]. This suggests that both auditory and visual imagery may be underpinned by common spatial mechanisms that allow for auditory images to be are extended in time akin to the way visual images are extended in space. Further, in behavioural tasks, auditory imagery can interfere or facilitate with the discrimination of auditory targets [27, 28] much like visual imagery can interfere or facilitate with the discrimination of visual targets [29, 30]. Similarly, in typical imagers, self-reported vividness of auditory and visual imagery are positively associated [10, 31] and the amount of grey matter volume in the Supplementary Motor Area (SMA) predicts individual differences in both domains [31].

Nevertheless, there is also evidence for modality-specific imagery processes. For example, primary and secondary association cortex are jointly activated by perception and imagery in the temporal and occipital cortices, respectively, for audition and vision [32, 33], although see [22, 34, 35]. Patient studies show that damage to brain regions relating to visual perception confer parallel impairments in visual imagery [36] and likewise damage to auditory perceptual areas confer impairments in auditory imagery [37], consistent with the suggestion that imagery engages modality specific perceptual systems but via a reverse hierarchy [38]. Given evidence for modality general and modality specific imagery processes, it would seem likely that imagery deficits could manifest across multiple modalities in some individuals and selectively in specific modalities in others, depending on the aetiology of the imagery impairment.

In summary, people with aphantasia often self-report deficits in multiple imagery modalities, including in the auditory domain. Previous studies have not tested whether these self-reported deficits in auditory imagery are associated with measurable deficits in performance on behavioural tasks that require auditory imagery. Here, we recruited a group of participants that self-identified as aphantasic and a matched control group of participants with typical imagery. We assessed how many of the participants with aphantasia, recruited on the basis of their reduced visual imagery, also reported concomitant auditory imagery deficits. We also tested their performance on musical pitch and voice-based tasks. We tested performance of the aphantasic group as a whole and also separately assessed only the subgroup that reported additional auditory imagery deficits. Given previous findings, we predicted that there would be a significant number of participants that self-reported both visual and auditory imagery deficits in the aphantasia group. Furthermore, we predicted that if these participants with aphantasia had deficits in auditory imagery then they would perform worse than participants with typical imagery on tasks that require auditory imagery, in line with their self-reported lack of auditory imagery.

## Materials and methods

### Participants

Twenty-nine individuals that self-identified as being aphantasic were recruited through social media platforms: the Aphantasia forum (*Aphantasia Forum*), two Aphantasia groups on

Facebook ('*Aphantasia non-imager/mental blindness awareness group*' and '*Aphantasia!*'), Twitter, and on aphantasia forums on Reddit. Data acquisition occurred between September 2019- March 2020, during which, aphantasia was defined specifically as the absence of visual imagery [2, 9] rather than an absence of sensory imagery [1]. We recruited participants on that basis to investigate the association of auditory and visual imagery deficits in individuals that would typically identify as being aphantasic (on the basis of their reduced visual imagery). This approach was taken to investigate the auditory imagery of individuals that would typically be identified as 'aphantasic' in previous research studies in this area that were collected during this time (between September 2019- March 2020).

Thirty control participants with typical imagery were recruited from University of Westminster campuses, the website *Call for Participants* and across social media platforms such as Twitter. At present, there is no agreed cut-off score for defining groups based on typical and atypical self-reported imagery. We used the VVIQ cut-off described in the methodology of our previous study [4] and one aphantasic participant was excluded as their score was above the cut-off. Aphantasic participants (VVIQ cut-off ≤ 25) comprised of 9 males and 20 females, mean age: 38y1m (SD = 11.33y). On the VVIQ, aphantasic participants scored a mean of 17.3 (SD = 2.51, minimum = 16, maximum = 24). Control participants with typical imagery (VVIQ cut-off ≥ 35) comprised 10 males and 20 females with mean age: 39y1m (SD = 11.02y). An independent samples t-test ($t(57) = 0.33$, p = .74, $d$ = .09) confirmed that there was no evidence of a difference in age between the groups. On the VVIQ, control participants with typical imagery scored an average of 62.03 (SD = 8.93, minimum = 41, maximum = 80) and were matched to aphantasic participants in terms of the region of the UK that they attended Primary School. This was to ensure a similar familiarity of regional accents between the groups for the voice identity imagery task. No participants were familiar with the voices of the speakers used in the voice task. The protocol for the study received ethical approval from the Research Ethics Committee of the Psychology Department at the University of Westminster, UK. Written consent was obtained from participants prior to taking part in the study.

## Bucknell Auditory Imagery Scale (BAIS)

The BAIS [39] is an auditory imagery questionnaire that comprises two scales: the vividness (BAIS-V) and control (BAIS-C) subscales. The vividness subscale requires participants to reflect on the vividness with which they can imagine 14 sounds described in a written scenario (e.g. "*consider attending a choir rehearsal, the sound of an all children's choir singing the first verse of a song*") on a scale of 1 *("no image present at all")* to 7 ("*as vivid as actual sound*"). These scenarios involve imagining different kinds of sounds that include voices and musical instruments. The control subscale comprises of 14 pairs of imagined sounds, which require participants to rate 'the ease of change' or transition on a scale of 1 *("no image present at all")* to 7 ("*extremely easy to change the image*") between two imagined auditory scenarios (e.g. *"Consider listening to a rain storm. The sound of gentle rain. The gentle rain turns into a violent thunderstorm").* Scores are calculated by the mean response to the 14 questions for each of the two subscales. In the current study we only report the BAIS-V due to a formatting error in the BAIS-C Likert scale that was presented to participants.

## Goldsmiths Musical Sophistication Index (Gold-MSI)

The Goldsmiths Musical Sophistication Index (Gold-MSI) is a measure of musical ability, preferences, and degree of sophistication of musical skill and behaviour [40]. This was included to measure the musical abilities of each participant group to allow us to account for any group differences in musical sophistication and musicality in the musical pitch task. The Gold-MSI

comprises of 5 stand-alone dimensions, each of which probe a specific aspect of musical behaviour such as (1) self-reported perceptual ability, (2) active musical engagement (i.e. amount of time spent undertaking music-related activities), (3) musical training, (4) self-reported singing ability and (5) sophisticated emotional engagement. The Gold-MSI also includes a sixth scale known as the General Musical Sophistication Scale, which encompasses all 5 of the dimensions. Questions included *"I can tell when people sing or play out of tune"* and *"I rarely listen to music as a main activity"*. The General Musical Sophistication Scale is composed of a total of 18 questions, and individuals are asked to rate on a Likert scale of 1 (complete disagreement) to 7 (complete agreement) how much they agree or disagree with each statement. Scores are entered into the Gold-MSI General Musical Sophistication scoring template to obtain a normalised value for each item, and these values are summed to provide an overall general musical sophistication score for each participant.

## Musical pitch imagery task

Musical imagery tasks, particularly tasks involving imagery of pitch, have been successfully used to investigate auditory imagery [23, 41–44]. People that self-report stronger auditory imagery have been shown to perform better on pitch imagery tasks [42]. One established task for assessing pitch imagery, requires participants to sustain an auditory representation of a well-known tune and to determine which of two target words/lyrics is pitched higher than the other [23, 37, 45]. Our task was based on the classic auditory imagery tasks described by [23] in which participants are required to compare the pitch of two imagined song lyrics. We also included a matched perceptual condition, involving pitch discrimination of heard tones equivalent to the intervals of the song lyrics in the imagery task. This additional condition was a control condition included to identify whether the groups differed on pitch perceptual abilities.

Twenty highly familiar songs were used and the title lyrics (e.g. "I wanna dance with somebody" by Whitney Houston) were used to create two alternative trials, generating a total of 40 trials, which were presented in two blocks of 20 trials each. These two trials derived from each song were used to compare the pitch of two different imagined syllables, e.g. "I **wan**na dance **with** somebody" and "I **wan**na dance with **some**body"). In 21 of the trials the second target lyric was higher, and in 19 trials it was lower in pitch, than the first. On each trial, the participant was presented with a lyrical phrase on the screen, along with the name of the artist. The two target lyrics where the respective pitch was to be compared were underlined, for example "I **wan**na dance **with** somebody" and participants were required to imagine the song in their head and to indicate by pressing a button on a keyboard whether the second underlined syllable was higher or lower in pitch than the first underlined syllable. Participants were also asked to rate the familiarity of all the songs prior to the imagery task on a scale of 1–5 (1 = *I do not know this song* to 5 = *I can hum or sing this song from start to finish*). If participants rated any songs either as 1 or 2, they were instructed to skip the trial on the associated lyric imagery task that followed. The experimenter stayed in the room (hidden from view) with the participant to ensure they did not hum or sing the lyrics.

A matching perceptual tone condition was created in which pure tones were synthesised to match the pitch of the target lyrics of the songs presented in the lyric imagery condition. We used pure tones to allow participants to focus on a simple pitch decision without additional distractions that might be generated by vocals and other musical features. On each trial, participants were asked to make a decision about whether the second tone that they heard was higher or lower than the first presented tone. Each stimulus was presented for 750ms with a 500ms silent gap between them. The perceptual condition comprised of two blocks of 20 trials. The sounds were presented via Sennheiser HD25 headphones.

## Voice task

Individuals with aphantasia report reduced facial recognition abilities [6, 9, 46]. The extent to which these facial recognition deficits are a consequence of poor visual imagery and their specificity to face rather than visual recognition more broadly is the subject of continuing research [47]. More broadly, the role of visual imagery in facial recognition remains unclear [48] and specifically whether being able to see someone's face in your mind's eye makes you better at recognising them [46]. Given that voices have been argued to function as "auditory faces" [49], we developed an equivalent voice recognition task. We hoped that this novel task would provide a complement to a more well established voluntary musical pitch imagery task.

As well as being able to imagine popular songs, most people are also able to recall an auditory image of the voice of a friend or a famous person. Voice identity categorisation can be investigated by morphing between the voices of different speakers. Using this approach, it is possible to linearly interpolate between two target voices, that reflect two distinct vocal identities, to generate a continua of sounds that transition in equal acoustic steps from one vocal identity to another, with novel ambiguous vocal identities generated at intermediate steps. Participants tend to sharply categorise continua of this kind into two distinct vocal identities, rather than in the continuous manner that might be expected given the incremental change in acoustic properties [50].

Categorical perception tasks of this kind have been used in many different contexts, including in colour, emotion, and phonology research, to probe the underlying structure of internal representations [51–53]. In this work, shallower categorisation functions are argued to indicate less well-specified or vivid internal representations [54–56]. To categorise a novel ambiguous vocal stimulus as belonging to a particular vocal identity, one may draw upon an internal representation of the to-be-categorised vocal identities to compare the novel auditory stimulus against. If individuals with aphantasia find generating vivid auditory images difficult, we hypothesised that they would display shallower categorisation functions, as they would summon less well-specified representations of vocal identities. Shallower functions would reflect greater uncertainty in labelling the vocal identities, in a similar manner to the way in which shallower identification functions are interpreted as reflecting less well specified phonological representations of speech sounds in those with language learning impairments [54]. In this way, we hoped that this task would allow us to index auditory imagery abilities without explicitly instructing participants to generate auditory imagery to avoid demand characteristics that might be associated with asking participants identified on the basis of reduced imagery to engage in an explicit imagery task.

In brief, pairs of the same Consonant Vowel (CV) syllables, e.g. /ka/, from different speakers were morphed, such that a continua was created that transformed from one speaker identity to the other, whilst maintaining the same linguistic percept, with intermediate vocal identities created in the midpoints of the continua. This was achieved in MATLAB 2017a (Mathworks, Inc., Natick, USA) using Speech Transformation and Reproduction by Adaptive Interpolation of weighted spectrogram (STRAIGHT) [57]. For each syllable, morphs were created through linear synthesis to create a continua with 11 steps (i.e. 0, 10, 20, 30, 40, 50, 60, 70, 80, 90, 100%), with each endpoint of the continua (i.e. 0 and 100%) pertaining to the original voice of each speaker. This morphing approach uses interpolation to create intermediate sound tokens between the two end point stimuli.

The percept of change of speaker that is expressed across the continua reflects changes across multiple simultaneously changing acoustic features. This approach provides highly natural sounding speech tokens but does not make it possible to identify how individual acoustic features are influencing speaker identity judgements. We used 4 continua to ensure that

multiple acoustic features would contribute to voice identification and provide a variety in speaker characteristics and phonetic content over which vocal characteristics were expressed. This helped to ensure that any between-group differences identified would be unlikely to be idiosyncratic to the specific set of speakers or speech sounds used. The syllables /ka/ and /ma/ syllables morphed between two different male speakers and the two syllables: /va/ and /ya/ were morphed between two different female speakers. The data from a third pair of speakers was collected but participants struggled to categorise the morphed continua into distinct vocal identities and so these data were not analysed further.

Prior to conducting the voice identify categorisation task, participants were trained to recognise and discriminate between the different speakers in order to develop an auditory image of the voice of each individual. To do this they first passively listened to 20 Bamford-Kowal-Bench (BKB) sentences. These are short 4–6 word sentences (e.g. *"they ate the lemon jelly")* that have a simple syntax and vocabulary [58]. They were spoken by each speaker (presented as 'Voice 1' and 'Voice 2') during an exposure phase. They were then actively trained to identify each of the two speakers using a further 40 novel BKB sentences (20 responses for Voice 1 and Voice 2) in which they heard the sentences and were required to categorise them as belonging to voice 1 or 2 whilst receiving corrective visual feedback. Following training, participants were tested on the morphed CV syllables which they had not heard during the exposure and training phases. They were required to categorise which of the two speakers they thought the spoken CV syllable belonged to. Crucially, as the CV syllables had not been heard by the participants before the participants needed to compare the ambiguous voices (at the intermediate parts of the continuum), that were a mixture of the two vocal identities, to an internal auditory image of the imagined vocal identities of the speakers. The continua were presented in a random order and each was repeated 12 times, resulting in a total of 528 experimental trials (11 continua steps x 4 continua x 12 repetitions per continua step). The pairs of speakers were presented one by one in a fixed order, e.g. the sounds from the male speakers and then the female speakers.

## Procedure

The study was undertaken in one single testing session (of approximately 90 minutes) based at the University of Westminster, UK. Each participant was tested individually. Participants first completed the VVIQ, Gold-MSI, and BAIS questionnaires followed by three computerised tasks that were presented in a randomised order: the voice task, the musical imagery pitch task, and a visual-spatial statement task (data not reported or included in the current manuscript). The imagery and perceptual conditions of the musical imagery pitch task were run consecutively, but the order of this was counterbalanced across participants.

## Statistical analysis

Data were analysed with R (https://www.r-project.org/). Anonymised data and R-Markdown documents detailing the analyses described below are available here: (https://osf.io/xgzct/). Parametric statistical tests were conducted when normality assumptions were met, and the data were transformed or the non-parametric equivalent statistical tests conducted when transformations were unsuccessful. Bayes Factors were calculated with JASP (https://jasp-stats.org/), to assess evidence in favour of the null hypothesis when statistical tests were not statistically significant. For these analysis we used the thresholds [59]: BF1 = "*No evidence*", BFs 1–3 = "*Anecdotal evidence*", BFs 3–10 = "*Moderate evidence*", BFs 10–30 = "*Strong evidence*", BFs 30–100 = "*Very strong evidence*", and BFs >100 = "*Extreme evidence*" to support the null hypothesis. BFs relative to the null model are reported for the Bayesian ANOVAs. Data

visualisations represent the raw not transformed data. All statistics analysed were evaluated based on a significance level of p < .05, and all p values are two-tailed.

## Results

The data were first analysed comparing all aphantasic participants to the control group of participants with typical imagery. A secondary analysis was then conducted only comparing the aphantasic participants who reported significant impairments in auditory imagery to the typical imagery control group.

### Self-reported auditory imagery

The data on the Bucknell Auditory Imagery Vividness scale (BAIS-V) were not normally distributed, which reflected a floor effect of the scores in the aphantasic group. Data transformations were unable to satisfactorily correct for this, so a non-parametric test was conducted. A Mann-Whitney test showed that controls with typical imagery (median = 5.5) scored significantly higher than participants with aphantasia (median = 1.1, W = 97.5, Z = 5.136, p < .001, Fig 1A).

Auditory imagery vividness scores for most aphantasic participants clustered below 2 out of 7, however, there were eight participants with scores above 2. Five of these participants had scores that were equal to or higher than the average of the typical imagery control group. This suggested that there might be a subgroup of participants with low self-reported visual imagery but typical auditory imagery. However, as this informal observation could equally represent sampling or measurement error, we formally tested this using the dip test [61], to assess whether the distribution of scores of the aphantasic participants on the BAIS-V was different to a unimodal distribution. This would be true if the scores were bimodal or multimodal, e.g. if they included a subgroup of participants with relatively high auditory imagery. There was no evidence to support this (D = 0.07, p = .324).

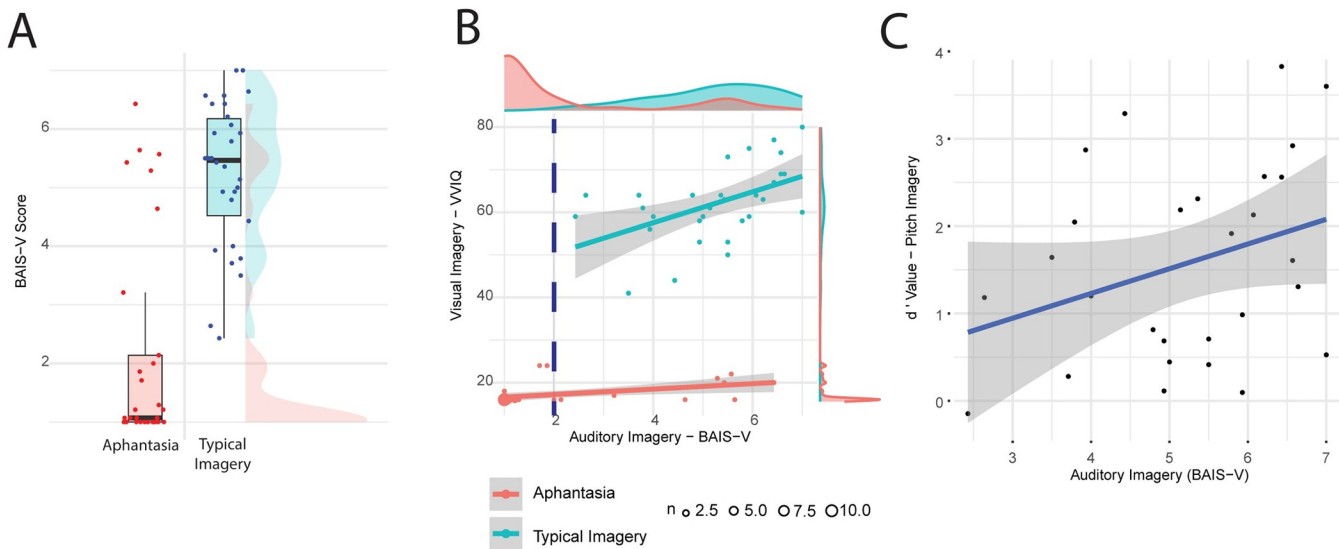

**Fig 1. Self-report measures.** (A) BAIS-V scores for each group, (B) the correlation between visual (VVIQ) and auditory imagery (BAIS-V) in each group and (C) the correlation between BAIS-V scores and the pitch imagery task in typical imagers. The blue-dotted line in (B) represents the BAIS-V value of 2 cut-off used to define aphantasic participants with low auditory imagery in a sub-analysis.

We further investigated the association between self-reported vividness of visual imagery (measured by the VVIQ) and vividness of auditory imagery (measured by the BAIS-V). A Spearman correlation showed that the vividness of auditory and visual imagery was correlated in both the aphantasic group ($r_s$ = .53, p = .003) and the typical imagery control group ($r_s$ = .53, p = .003, Fig 1B). There was no evidence of a difference in the strength of association between auditory and visual imagery, when the correlation coefficients were directly compared between groups following conversion to Fisher Z values (p = .984). This was consistent with the previous finding suggesting the absence of clear evidence of a subgroup of participants with aphantasia with low visual imagery but high auditory imagery. Hence, whilst imagery was lower in the aphantasic group, there was a similar relationship between imagery scores expressed across modalities in the typical imagery control and aphantasic groups.

## Musical pitch imagery task

Participants were excluded from the analysis if they reported being unfamiliar with more than 5 songs (operationalised as a score < 3 in familiarity over a quarter of the trials). This resulted in the exclusion of two control participants with typical imagery (both of whom reported 8 of the 20 songs in the study as unfamiliar), leaving 28 control participants. Three aphantasic participants were removed on this basis (these 3 participants on average reported 9 of the 20 songs as unfamiliar), resulting in 26 aphantasic participants. These participants were also excluded from the analysis of the tone perception condition but were not removed from the voice task analysis.

Performance in the two conditions was compared between aphantasic and control participants with typical imagery using d-prime. A mixed 2 x 2 ANOVA with factors participant group (aphantasic / typical imagery control) and task (lyric imagery / perceptual tone) showed that there was a significant main effect of task ($F$(1, 52) = 145.54, p < .001, $\eta p^2$ = .74, Fig 2A) with participants performing better in the perceptual condition (d-prime: M = 3.22, SD = 0.67) than the imagery condition (d-prime: M = 1.53, SD = 1.03). There was no main effect of participant group ($F$(1, 52) = 0.20, p = .656, $\eta p^2$ = .004, $BF_{01}$ = 4.18) and critically no significant interaction between task and group which would have indicated a selective deficit in pitch imagery in the aphantasic participants ($F$(1, 52) = 1.65, p = .205, $\eta p^2$ = .03, $BF_{01}$ = 1.73). These results show that despite self-reporting a lack of auditory imagery, participants with aphantasia did not significantly differ from participants with typical imagery on this task.

In order to account for any differences in musical sophistication or familiarity of the songs used in the musical imagery task, we tested for a difference between groups on each of these measures. An independent samples t-test showed that aphantasic participants scored lower (mean = 54.8) in musical sophistication, as measured by the Goldsmiths Musical Sophistication Index, than the typical imagery control group (mean = 75.5; t (57) = 2.90, p = 0.005, d = 0.76). Consistent with this, the aphantasic group (mean = 3.36) also reported being less familiar with the songs used in the musical imagery task compared to the typical imagery control group (mean = 3.68; t (57) = 2.383, p = .021, d = 0.62). Hence, re-ran this analysis as a 2 x 2 mixed ANCOVA with factors participant group (aphantasic / typical imagery control) and task (lyric imagery / perceptual tone) with Gold-MSI and familiarity ratings as covariates. This again showed that there was no evidence of a significant interaction between task and group ($F$(1, 50) = 0.01, p = .912, $\eta p^2$ = < .001, $BF_{01}$ = 1.84) indicating that the aphantasic participants did not show evidence of a selective deficit in pitch imagery, when accounting for musical sophistication and song familiarity.

To further understand the implications of these findings, we further correlated the auditory and visual imagery scores with d-prime scores on the pitch task in the typical imagery group.

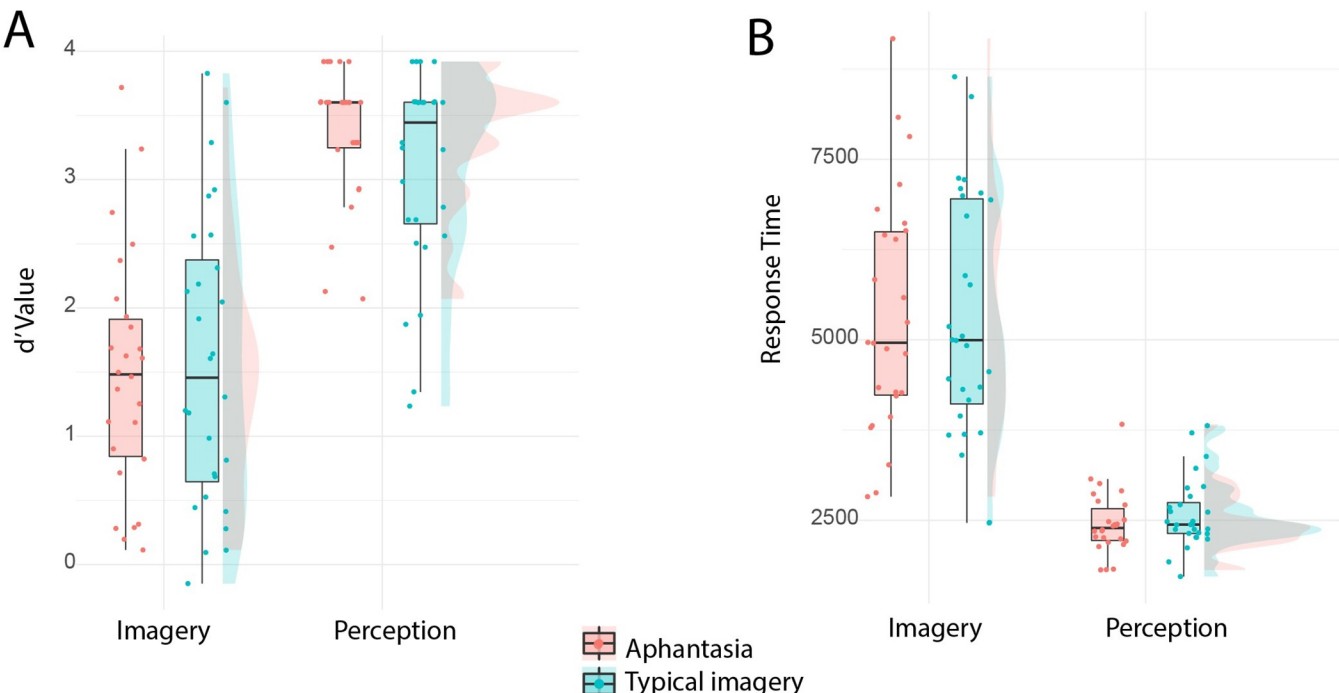

**Fig 2. Pitch Imagery task.** (A) d-prime values and (B) Response time (msecs) for participants with aphantasia and those with typical imagery in the musical imagery task.

We used a one-tailed test as we assumed that the relationship would be positive, e.g. increased self-reported imagery would be associated with better performance on the pitch task, and we had no basis to hypothesize a negative relationship. This indicated that there was a weak correlation between the auditory imagery scores (BAIS-V) and performance on the pitch task ($r = 0.32$, $p = 0.049$, Fig 1C) and a marginal relationship between the visual imagery scores (VVIQ) and performance on the pitch task ($r_s = 0.31$, $p = 0.052$). We further generated a composite imagery score by z-scoring the auditory and visual imagery scores and averaging them. This imagery composite score also correlated weakly with performance on the pitch task ($r = 0.33$, $p = 0.045$). From this we concluded that our pitch task indexed voluntary imagery ability, but did not distinguish between participants with aphantasia and those with typical imagery.

Equivalent analyses were run using response time as a dependent variable, as previous studies have shown evidence for differences in response time but not accuracy in some tasks in aphantasic individuals [4]. The response times were transformed using a Box-Cox transformation to meet the normality assumptions of ANOVA [60]. A mixed 2 x 2 ANOVA with factors participant group (aphantasic / typical imagery control) and task (lyric imagery / perceptual tone) showed a significant main effect of task ($F(1, 52) = 363.97$, $p = < .001$, $\eta p^2 = .88$, Fig 2B), showing that participants were significantly quicker in the perceptual tone condition (M = 2.52 seconds, SD = 0.46) compared to the lyric imagery condition (M = 5.32 seconds, SD = 1.65). There was no significant main effect of participant group ($F(1, 52) = 0.16$, $p = .688$, $\eta p^2 = 0.003$, $BF_{01} = 4.40$) or significant interaction between task and group ($F(1, 52) = 1.04$, $p = .312$, $\eta p^2 = .020$, $BF_{01} = 2.49$), indicating that participants with aphantasia were not significantly slower on the imagery task.

To account for differences in musical sophistication and song familiarity between the groups, we further conducted a 2 x 2 mixed ANCOVA with factors participant group

(aphantasic / typical imagery control) and task (lyric imagery / perceptual tone) with Gold-MSI and familiarity ratings as covariates. This showed that there was no evidence of a significant interaction between task and group ($F$ (1, 50) = 0.002, p = .968, $\eta p^2 < 0.001$, $BF_{01}$ = 2.32) suggesting that participants with aphantasia were not specifically slower in the imagery task when accounting for differences in musical sophistication and familiarity.

Taken together, these results suggested that participants with aphantasia did not show specific deficits in imagery measured by response time or d-prime values, compared to those with typical imagery on the musical pitch task.

## Voice task

In the training phase of the task, both participant groups were highly accurate at perceptually discriminating the two different speakers following the exposure phase (all > 99% for each speaker pair) indicating that they were able to accurately identify and learn the two different vocal identities following training. In the testing phase data, logistic regression functions were fitted to each participant's categorization function. The β value of the slopes were extracted as an indicator of the slope of the categorization function, as a reflection of how categorically each participant labelled the sounds from the continua as belonging to one or the other speaker. The assumption being that individuals that were able to generate a vivid auditory image of a speaker's voice would show a sharper categorisation, e.g. they would be more confident in assigning the novel sounds to one identity or the other. A larger β value represents a more categorical internal auditory representation of the two speakers' voices. One aphantasic participant was removed from the analyses as they confused which speaker was which on two of the continua. It was clear from the data that they systematically mislabelled the voices, e.g. assigned speaker 1 to speaker 2 and vice versa. No other participants were excluded in the analysis.

The slope parameter values were winsorized, recoding all winsorized values above the 95th to the value attained at the 95% percentile. This was necessary to account for extreme values arising in a small number of participants with exceptionally categorical responses that perfectly categorised the speakers for particular continuum sets. Following winsorization, slope parameters were Box-Cox transformed to approximate normality and submitted to a 2x2 mixed ANOVA with Greenhouse Geisser correction with factors participant group (aphantasic / typical imagery control) and continua (1, 2, 3, 4). This showed that there was a significant main effect of continua ($F$(2.24, 125.33) = 3.84, p = .02, $\eta p^2$ = .06, Fig 3A and 3B). This reflected the fact that there were differences in the sharpness of categorisation across continua. However, there was no main effect of participant group ($F$(1, 56) = 1.473, p = .23, $\eta p^2$ = .03, $BF_{01}$ = 1.84) or interaction ($F$(2.24, 125.33) = 1.57, p = .21, $\eta p^2$ = .03, $BF_{01}$ = 3.81). These results indicate that there was no evidence of a difference between aphantasic participants and those with typical imagery in their ability to generate vivid auditory representations of different speakers.

To provide further context to this finding, we correlated the slope parameter values averaged across continua separately with the visual (VVIQ) and auditory imagery (BAIS-V) scores in turn. There was no evidence of a relationship between auditory or visual imagery self-report and performance on the voice task alone or when the imagery measures were combined in a composite score by z-scoring and then averaging them (all ps > 0.575). We then directly tested for a difference in the strength of correlation between the auditory imagery values and the pitch and voice tasks respectively. This showed that the pitch imagery task was more strongly correlated with auditory imagery scores (BAIS-V) than the voice task was (z = 2.20, p = 0.028).

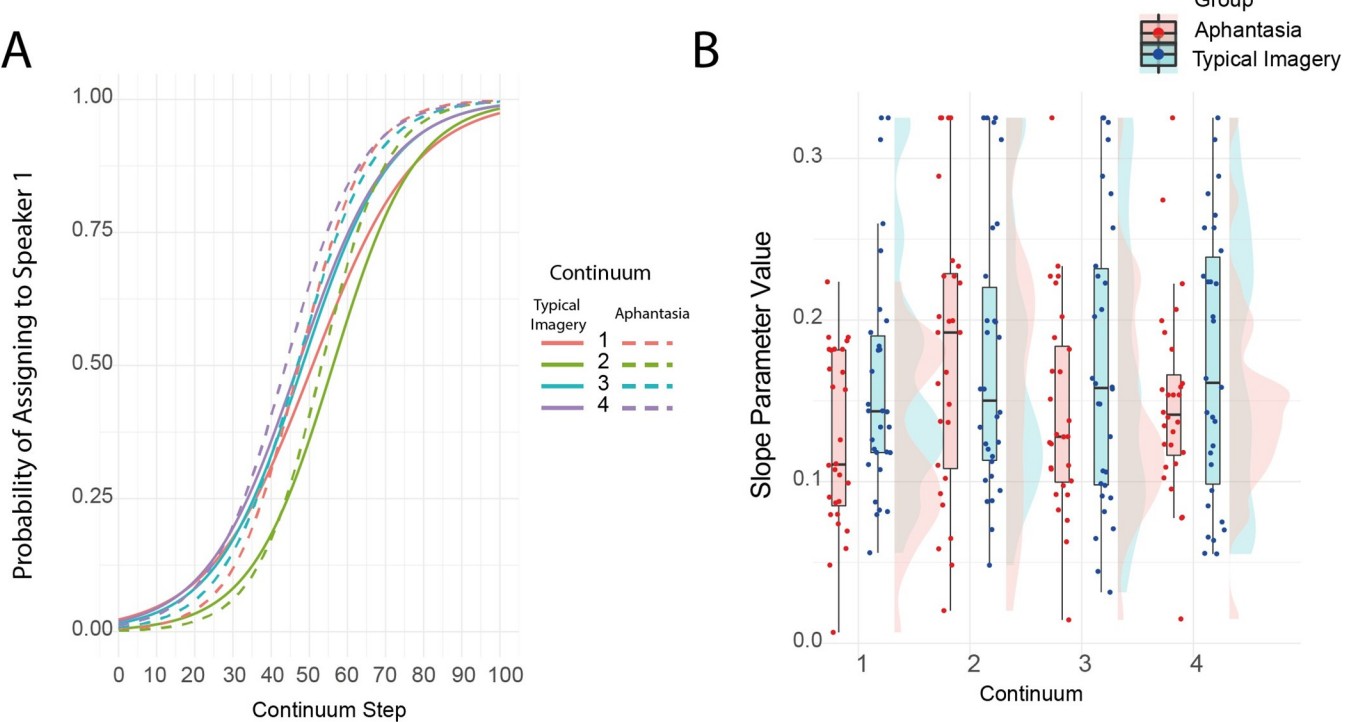

**Fig 3. Voice task.** (A) Averaged logistic regression functions and (B) slope parameter estimates displayed by group and continua.

### Further comparison of aphantasia and control groups

As the reader will recall, we did not find statistical evidence, with the dip test, [61] for a subgroup of aphantasic participants with intact auditory imagery abilities. However, we decided to conduct an additional analysis only including participants with aphantasia with the lowest self-reported auditory imagery. We did this for completeness to ensure that the previous null findings could not be explained by the inclusion of the participants with better auditory imagery in the aphantasic group. The subgroup was selected by excluding the eight participants with aphantasia with a mean BAIS-V score greater than 2 (M = 4.79, SD = 1.43, minimum = 2.14, maximum = 6.43). This left a sample of 21 participants with aphantasia, all of whom reported low auditory imagery. The remaining sample included 8 males and 13 females, mean age: 39y09 (SD = 11.81), mean VVIQ score: 16.86 (minimum = 16, maximum = 24, SD = 2.41) and BAIS-V score: 1.17 (minimum = 1, maximum = 2, SD = 0.30). When all the equivalent analyses were conducted with this subgroup of participants with the lowest self-reported auditory scores, the analyses were unchanged. There was no evidence of a difference between the groups on either the musical pitch or voice tasks. Taken together we could find no clear evidence for differences in auditory imagery abilities in individuals that self-identify as aphantasic or more specifically in those who self-report auditory imagery deficits.

### Discussion

The current study compared the performance of individuals with aphantasia to participants with typical imagery on behavioural tasks we thought would require auditory imagery. As predicted, at a group level, there was a significant co-occurrence of self-reported visual and auditory imagery deficits. Indeed, the majority of aphantasic participants clearly reported deficits

in both auditory and visual imagery (n = 21, 72% when defined as a VVIQ score below 25 and an average BAIS-V score below 2). Despite profound differences in self-reported auditory imagery experience between groups, we did not find evidence of reduced performance or increased response times on the auditory imagery tasks, either when considering the full sample of participants with aphantasia or when only considering those with the lowest reported auditory imagery. This represents a significant discrepancy between the subjective experience of auditory imagery and our ability to measure these deficits in objective auditory tasks. These findings are discussed in relation to the mechanisms that might act to obscure imagery deficits in auditory imagery tasks and potential limitations of our methodology.

The majority of aphantasic participants in our sample that self-reported low visual imagery also reported reduced auditory imagery. This is consistent with previous studies of individuals with aphantasia that have shown reduced self-reported imagery across multiple sensory domains [7–9]. We found significant correlations between self-reported auditory and visual imagery in both the typical imagery control and aphantasic groups, suggesting that introspection concerning the vividness of sensory imagery shares modality general processes [21] and is consistent with previous evidence showing a high concordance between self-reported auditory and visual imagery abilities [31]. We did not find evidence for a difference in the strength of correlation between reported imagery ability across modalities comparing between the groups. This suggests that the participants with aphantasia in our sample had weaker but not qualitatively different imagery processes compared to those with typical imagery, albeit within the constraint that this conclusion is drawn from self-report data only.

It should be noted that there were some aphantasic participants that reported low visual imagery but relatively high auditory imagery (8 participants with BAIS-V > 2, of which five participants had values equal to or above the average for the typical imagery group). Disassociations across sensory domains is not uncommon in many clinical conditions, for example patients with conditions (e.g. hemi-spatial neglect), can show neglect that is comorbid with or arise in only one sensory domain [62]. The individual differences we observed in auditory imagery abilities in participants with aphantasia could be suggestive of a disassociation in imagery experience across modalities in a small number of individuals. We investigated this empirically by testing to see if there was evidence that the distribution of auditory imagery scores in the aphantasic participants differed from a unimodal distribution (e.g., whether it reflected a sub-group with good auditory imagery and another with poor auditory imagery) and did not find evidence to support this. This is also consistent with the lack of evidence for a difference in the strength of association between visual and auditory imagery between the aphantasic and typical imagery control groups.

Drawing this together, we did not find evidence for a subgroup of aphantasic participants with low reported visual imagery but relatively intact auditory imagery. As with any statistical test, we may not have had sufficient power to identify a sub-group. This might also explain our failure to find differences between groups in the auditory imagery tasks more broadly. However, in mitigation, we note that our sample size was consistent with many previous in-person studies of individuals with aphantasia [5, 6]. Recruiting individuals with aphantasia for in-person research can be difficult due to the low incidence within the population [63]. Future studies using online behavioural experiments [64] may help to address this issue, provided that data quality can be assured. More broadly, this study indicates that further empirical studies are required, supported by both self-report and behavioural observation on tasks that require imagery, to verify and support the utility of aphantasia subgroups [1, 11, 63].

The discrepancy we observed between participant report and behavioural performance on tasks involving imagery is not specific to the auditory domain as other studies within the visual domain have found a similar discrepancy [4, 6]. There may be several reasons as to why the

differences are not evident in the current study. One explanation is that some individuals with aphantasia may have unconscious auditory imagery [19] and are able to implicitly use auditory imagery but have no explicit conscious awareness of this auditory experience. Alternatively, it may be that differences only arise on self-report because the two groups differ in their thresholds for labelling an image as 'vivid', in that individuals with aphantasia have a much higher threshold for reporting an image as vivid compared to those with typical imagery [65] with these thresholds potentially also differing across modalities. Equally, our findings might support the broader assertion that the experiential phenomenology of imagery is epiphenomenal and does not play a causal or functional role in cognition [16]. However, we note that these explanations cannot account for previous findings showing positive differences on behavioural visual imagery and visual imagery related tasks in people with aphantasia [3, 5].

Equally, as has been suggested in the visual domain, individuals with aphantasia might use non-visual strategies to compensate for their imagery deficits (e.g. spatial or verbal strategies). In the current study, it is possible that spatial strategies may have been used to aid performance. Voices and melodies both comprise pitch information [66, 67]. Pitch is often described using spatial terms and is organised on a scale of low to high [68] giving rise to the concept of 'pitch height' along a vertical location which may facilitate re-mapping of pitch to a spatial representation [69]. For instance, pitches that are 'high' are perceived to be situated in higher visuospatial location (e.g. 'up' in space) compared to lower pitches [70, 71], with a similar mapping existing between auditory pitch and tactile location [72].

Evidence from individuals with amusia suggests that pitch processing may depend on the same cognitive mechanisms that are used to process spatial representations in other modalities [73, 74]. For instance, individuals with amusia have difficulties discriminating changes in pitch and perform worse on spatial tasks such as mental rotation compared to neurotypical individuals [73, 74]. Given that aphantasic individuals self-report and show intact spatial imagery abilities in the visual domain [3, 5, 8], it could be that the spatial equivalent is intact and exploited within the auditory domain to perform these auditory imagery tasks. Indeed, they may have been able to exploit this in both the musical pitch and voice task, as they may have used pitch information to represent the vocal identities. For instance, the voice task was based on pairs of voices that were relatively distinctive in their vocal qualities; thus it did not necessarily rely on the need to build rich auditory representations of the voices, but merely to have a low-level and 'non-pictorial' understanding of the differences between voices (e.g. one voice being higher pitch than other), from which to base subsequent responses. Conversely, in the musical imagery pitch task, while the experimenter was present in the room to ensure participants did not hum or sing, they could have mapped this through touch, for example, made small sweeping motions (unnoticed by the experimenter) with their finger in order to aid with spatial mappings (indeed, there are wide variations amongst individuals between the sensory attributes and features that are associated with these mappings, see [72, 75].

It may have been the case that our tasks were not sensitive to the auditory imagery deficits exhibited by individuals with aphantasia. We chose two tasks. One was a traditional voluntary pitch-based imagery task; performance on this task was correlated with self-reported auditory imagery vividness but it did not identify a difference in performance between groups. Given that this task explicitly asked aphantasic participants to generate auditory imagery, it may have been subject to demand characteristics. Further, all questionnaires were undertaken prior to undertaking the task, which may have also suggested to participants that imagery was necessary for task performance. However, one might assume that such a bias would lead to reduced rather than increased performance in the aphantasic group. Given that performance was equivalent between the groups, it may be that participants with aphantasia drew upon alternative non-imagery-based strategies to complete the tasks [4].

Our voice task was potentially less susceptible to demand characteristics because we did not explicitly instruct participants to engage in auditory imagery during the task. We did not find evidence that performance on this task was correlated with auditory imagery self-report and as such it may not have been an effective task for measuring auditory imagery ability. This may be because participants were able to identify and use acoustic features as verbal labels to differentiate between speakers without needing to generate an auditory image. However, the absence of differences between the groups on this task is interesting, especially given recent findings of facial recognition deficits in individuals with aphantasia [6, 9], albeit these deficits may have been driven by a more general visual deficit [47, but also 4]. Here, in the auditory domain we did not find evidence of an equivalent deficit for voice recognition. Further research is required examining the relationship between perception and imagery across a range of tasks and modalities in individuals with aphantasia.

Given our failure to find differences between the groups, what alternative approaches or tasks might be better suited to identifying auditory imagery deficits in those with aphantasia? In the visual domain, the most consistent evidence of imagery deficits in individuals with aphantasia comes from tasks in which participants are shown to exhibit a reduced perceptual bias or facilitation via imagery priming prior to a perceptual decision [5, 17]. Future attempts to find behavioural corroboration for the reported auditory imagery deficits may benefit from taking a similar imagery priming approach in the auditory domain [76].

Another reason for the failure to show group difference on the tasks may be due to our sampling approach. We recruited participants on the basis of visual imagery deficits; this may have biased us towards testing participants with less extreme auditory imagery deficits than if we had specifically recruited participants on the basis of reduced auditory imagery. However, this seems unlikely given that auditory imagery scores were so low in our sample even when including those participants with higher BAIS scores (a median of 1 out of 7, "no image at all"). Future studies may benefit from using an objective criterion, rather than self-identification, as a basis for defining an aphantasia group. Indeed, a number of recent studies have identified physiological bases by which individuals with aphantasia may be identified [14, 15]. Some research groups have used these objective measures to identify their aphantasic participants [15]. This approach shows promise; however, it requires agreement within the field as to the most appropriate technique to use for this purpose. We hope that this initial work assessing performance on auditory imagery tasks in individuals with aphantasia provides a basis for future investigations of the objective imagery behaviour of those with aphantasia in other imagery domains.

## Conclusions

Recent research shows that individuals with aphantasia report impairments in imagery in domains beyond vision, including in audition. However, until now, there has been little work to establish behavioural corroboration of these reported deficits. In a group of participants that self-identify as being aphantasic and that report both visual and auditory imagery deficits, we were unable to find evidence of reduced performance on a musical and voice-based imagery task. The reason that this was the case remains unclear. Further research is required to verify and support the utility of identifying subgroups of people with aphantasia on the basis of the pattern of imagery deficits across modalities in both self–report and imagery behaviour. We hope that this initial work in this area provides a basis for future investigations of the objective imagery behaviour in those with aphantasia in other imagery domains.

## Acknowledgments

With thanks to all participants who generously offered their time to participate within this study. An additional thanks to Jeff Knowler for his assistance in creating the tones within the musical lyric perceptual condition, to Dr Amy Woy for providing the singing clips for the practice trials of the musical lyric imagery condition and to the speakers who kindly gave time to record their voices for the voice task.

## Author Contributions

**Conceptualization:** Zoë Pounder, Alison F. Eardley, Catherine Loveday, Samuel Evans.

**Data curation:** Zoë Pounder.

**Formal analysis:** Zoë Pounder, Samuel Evans.

**Methodology:** Zoë Pounder, Alison F. Eardley, Catherine Loveday, Samuel Evans.

**Project administration:** Zoë Pounder.

**Resources:** Zoë Pounder, Samuel Evans.

**Software:** Zoë Pounder, Samuel Evans.

**Supervision:** Alison F. Eardley, Catherine Loveday, Samuel Evans.

**Visualization:** Zoë Pounder.

**Writing – original draft:** Zoë Pounder.

**Writing – review & editing:** Zoë Pounder, Alison F. Eardley, Catherine Loveday, Samuel Evans.

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
