## [Decision Letter · Decision Letter 0]

2 May 2023

PONE-D-23-05751No clear evidence of a difference between individuals who self-report an absence of auditory imagery and typical imagers on auditory imagery tasksPLOS ONE

Dear Dr. Evans,

Thank you for submitting your manuscript to PLOS ONE. After careful consideration, we feel that it has merit but does not fully meet PLOS ONE’s publication criteria as it currently stands. Therefore, we invite you to submit a revised version of the manuscript that addresses the points raised during the review process.

We look forward to receiving your revised manuscript.

Kind regards,

Jie Wang, Ph.D.

Academic Editor

PLOS ONE

Journal Requirements:

2. Please ensure that you refer to Figure 1-4 in your text as, if accepted, production will need this reference to link the reader to the figure.

Additional Editor Comments:

The authors should pay careful attention to each of the comments below and address the issues raised by the two reviewers.

Reviewers' comments:

Reviewer's Responses to Questions

**Comments to the Author**

1. Is the manuscript technically sound, and do the data support the conclusions?

Reviewer #1: Yes

Reviewer #2: No

2. Has the statistical analysis been performed appropriately and rigorously? 

Reviewer #1: Yes

Reviewer #2: Yes

3. Have the authors made all data underlying the findings in their manuscript fully available?

Reviewer #1: Yes

Reviewer #2: Yes

4. Is the manuscript presented in an intelligible fashion and written in standard English?

Reviewer #1: Yes

Reviewer #2: Yes

5. Review Comments to the Author

Reviewer #1: In this interesting and well executed study, the authors investigate whether participants who would qualify as aphantasic – an inability to generate mental imagery - based on a questionnaire measuring their visual imagery vividness (the VVIQ) also show reduced auditory imagery on both questionnaire measurements and objective task performance. To this end, 29 aphantasic (VVIQ < 25) and 30 control (VVIQ > 35) participants filled in questionnaires regarding their auditory imagery and musical sophistication and performed a musical imagery and a voice imagery task. The experiments and analyses are well-executed and controlled, the study investigates an interesting question that will increase our understanding of the cognitive mechanisms underlying aphantasia and caveats and alternative hypotheses are acknowledged. I only have two major comment that needs to be addressed.

Firstly, I want to applaud the authors for sharing the data and code as R markdown files that were available during review. This is a great example of open science!

The voice imagery task is interesting and well designed; however, I am not entirely sure to what extent it requires auditory imagery. The authors argue ‘as the CV syllables had not been heard by participants before, the participants needed to compare ambiguous voices … to an internal auditory image of the imagined vocal identities of the speaker’. However, this can be seen as a straightforward perceptual categorization task with new instances – the fact that these instances cannot be remembered but need to be compared against a category does not necessarily mean that imagery is needed for this process, right? Would the authors suggest that imagery is needed for all perceptual categorization of new instances? This needs to be discussed in more detail either in the methods section or in the discussion section as a possible pitfall of the study.

Is it possible that response biases play a role in the results such that self-identifying aphants just have a higher criterion for saying that their imagery is vivid compared to others, but in fact experience the same thing? This would explain the cross-modality correlation in questionnaire scores as well as the absence of objective behavioural effects. The authors suggest the aphantasia group might not have conscious imagery and that could explain the discrepancy, but this would be a slightly different mechanism; i.e. they might experience it the same way, but are less likely to say it is vivid.

Minor comments:

Was there a correlation (across groups) between questionnaire measures of imagery and task performance? If not, what does this mean about the validity of these tasks for testing imagery?

N = 29 is way too small to say anything about proportions that generalize to the population.

Line 243, I am not entirely sure I understand the perceptual control task; was new sound created for this task? Why was not just the original song used as control?

Line 294, what are ‘CV syllables’?

Was there a correlation between the control and vividness subscales of the auditory tasks and between the control of auditory imagery and vividness of visual imagery?

Reviewer #2: *** beginning of review

General Issues:

A. Epiphenomenon. The authors offer speculation why there might not be a difference in performance on auditory imagery tasks between participants who self-identified as having aphantasia and matched control participants (e.g., lines 101-109), but one explanation that the authors did not discuss is that imagery is not causal in the imagery tasks but is simply an epiphenomenon of the mechanisms that are causal. An analogy from study of visual imagery (which I think is due to Pylyshyn, but I’m not sure of the source) involves the light on a desktop computer that begins blinking when the computer is booting up. That blinking light plays no causal role in the functioning the computer, and the computer could complete booting-up just as well if that light was burned out. In the case of imagery, images are like that blinking light in that they are just a byproduct of the booting up process and play no causal role in that process. I personally am not convinced by previous arguments in the literature that imagery is epiphenomenal and does not play a causal role in cognition, but the author’s data can be interpreted as supporting that position. This epiphenomenal alternative should be discussed in the manuscript.

B. Thresholds. Another alternative that should be addressed is the possibility that the participants who self-reported aphantasia and the matched control participants might have equally vivid levels of imagery, but the participants who self-reported aphantasia have a much higher threshold for labeling an image as vivid. In other words, an image of a given level of vividness might be labeled as “strong” or “vivid” by person who does not self-identify as having aphantasia but as “weak” and “not vivid” by a person who does self-identify as having aphantasia. This possibility occurred to me at a couple of different locations in the manuscript (e.g., lines 364-383, 412-414, 427-429, 521-543) and should also be discussed in the manuscript.

C. Task Administration. The authors provide reasonable descriptions of the different experimental tasks. However, information regarding the experimental procedures and the administration of the experimental tasks is missing. In what order were the tasks given? Was order counterbalanced or randomized, and if not, then why not? How long did the completion of all the tasks require? Was this all done in a single session for each participant? Were participants always run one at a time, or were they run in groups? Etc.

D. Demand characteristics. Through its history, imagery research has always struggled with the issue of demand characteristics (i.e., the possibility that participants can deduce the experimental hypothesis and then modify their responses to reflect what they think the experimenter wants to observe). Given that participants who self-reported aphantasia could presumably deduce the authors’ interests in aphantasia (after all, those participants were recruited from aphantasia websites), what precautions were taken to limit any possible role of demand characteristics or to assess whether demand characteristics contributed to the data? This needs to be discussed.

E. Continua. I’m not clear regarding the nature of the continua that underlay the morphing of speaker (frequency? timbre? voice onset time? just a mix of “whatever”? etc.) There are numerous possible dimensions along which speech stimuli (or auditory stimuli more broadly) can differ, and if the effect of morphing is going to be related to imagery, readers need to know what underlying dimensions of the stimuli are involved. This becomes more important given that there has been some debate regarding which properties of auditory imagery are necessary and which are optional (e.g., Intons-Peterson has previously argued that loudness is not a necessary property of auditory imagery). Also, there might have been different features changed in different continua, and if so, then the dimensions along which change occurred might be confounded. This needs to be discussed.

F. Parallel Construction. There are a few places in the manuscript in which the authors need to use parallel construction. For example, the authors refer to “impairments” in line 61 but “experiences” in line 63, but I think they also meant “impairments” in line 63. In another example, they refer to “individuals” in page 515 and “participants” in line 516, but they should use the same term in both places. In general, using multiple terms for the same idea, construct, or entity makes more work for the reader, and so terminology should be more consistent across the manuscript. I’ve pointed out two instances, but the authors should check the document for other instances. On a related note, there are inconsistencies in the way that references within parenthetical reference lists are organized. The organization sometimes appears alphabetical by first author and not chronological (lines 58-59; 68-69), chronological but not alphabetical by author (e.g., line 89) or neither chronological nor alphabetical (e.g., lines 97-98). This should be more consistent throughout the manuscript.

G. Figures. Figures 1, 2, and 4 are very faint in my printed copy of the manuscript. I would suggest using a higher contrast and more saturation. There does not appear to be a Figure 3 in my copy of the manuscript, and I don’t know if this is a simple labeling error or if an intended figure was actually omitted.

H. Paragraph lengths. There are several paragraphs that are too long (.e.g., lines 85-113, 114-146, 544-573, 574-601). Breaking these into 2-3 smaller paragraphs (and adjusting the organization as needed) would make this easier on the reader.

I. References. There are a number of problems with the References. Pounder et al. (2021) is not in the References section, although there is a Pounder et al. (2022) that might have been meant. Similarly, Banno et al. is listed as (2006) in the main text but (2007) in the References section. Crowder et al. (1989), Jacobs et al. (2018), Keogh and Pearson (2021), Knauff et al. (2000) are cited in the main text but are not listed in the References section. Furthermore, Alderson-Day and Fernyhough (2015), Berry and Laskey (2012), Bench et al. (1979), Brogaard and Gatzia (2017), Emerson and Laskey (2012), Fegen et al. (2015), Floridou et al. (2015), Goggin et al. (1991), Holmes et al. (2004), Keogh et al. (2021), Lavan et al. (2019), Perrachione et al. (2011), Perrachione et al. (2009), Pounder et al. (2022), Price (2012), Scott and McGettigan (2015), Shergill et al. (2001), Vilhauer (2016), and Williamson et al. (2012) are all listed in the References section but are not cited in the main text. There are problems with individual citations as well (e.g., formatting of Box and Cox, 1994; Crowder, 1989; spelling in Dawes et al., 2022; Lui and Bartolomeo is listed as “2023” in the text but as “under review” in the text). Zatorre and Halpern (1993) is listed twice. I am sorry of this sounds overly harsh, but I think the authors need to know that when any reviewer sees this many problems with the References involving issues that should have been checked and corrected before submission, it implies to that reviewer that the authors haven’t done due diligence and were not sufficiently careful in preparing the manuscript (i.e., if the authors appeared this careless with their references, a reviewer would likely question whether the authors were sufficiently careful in preparing the rest of the manuscript [including the details of method, results, and analyses]).

Specific Issues:

Line 44: I would delete the comma after “controls”. More importantly, the phrasing suggests the same individual had to self-identify as both having aphantasia and as a matched control, which obviously isn’t what the authors intended,

Lines 69-71: Why is this claimed to be heterogeneous? If the same patterns are occurring across modalities, wouldn’t that suggest a more homogenous (or similar) mechanism? Am I missing something? Also, what does it mean to report “associations” in imagery deficits? Are the authors referring to something like a difficulty in imaging visual height is associated with a difficulty in imaging auditory pitch (cf. lines 586-595)? It seems like it would be easier to just say something like “report related deficits in multiple modalities.”

Lines 96-97. What the authors appear to be saying here is that participants who self-reported aphantasia can, with extra time, perform just as well as normal imagers unless the task is particularly difficult. It might be worth explicitly saying that. Also, this finding might offer evidence against the epiphenomenal view of imagery (see General Issue A).

Line 111: “the experience of people with aphantasia...” Isn’t the point that people who self-report having aphantasia don’t have an experience (by definition)? Or are the authors referring to a potential non-imaginal experience?

Lines 134-136: Can “primary sensory… respectively” be stated more simply? It won’t be clear for many readers.

Line 147: The antecedent of “other” is not clear. Also, “people with aphantasia” would seem to include all imagery modalities, so what would be “other” than all imagery modalities? The authors probably meant something like “multiple” or maybe “visual”, but that isn’t what they actually say.

Lines 151-154: This phrasing implies the authors are expecting the matched control group to also report deficits. Also, the phrasing implies the participants tested themselves.

Lines 157, 549: Do the authors really want to refer to aphantasia as a comorbidity or a morbidity, especially in light of the fact that aphantasia doesn’t seem to hinder normal functioning?

Lines 191, 215, 219, 279: This is a misuse of the term “paradigm”, although I admit it is a misuse that appears common in the psychological literature. A “paradigm” refers to the axioms, methods of investigation, and appropriate topics of investigation underlying a broad area of study. Examples of paradigms within psychology include behaviorism, information processing, and connectionism (neural networks). The term “paradigm” does not apply to the methodology or design of a single experiment or small set of experiments.

Line 205: Is this correct? Why does an anchor term for the control subscale refer to the vividness of the imagery rather than to the ease with which the initial image could be transformed into another image (i.e., controlled)? If this is correct, it would seem to offer a significant challenge to the validity of the BAIS-C subscale.

Lines 224, 293, etc.: As a reader, I never like it when I am directed to another source to find methodological and other information that is important for evaluating the article I am reading. I might be reading in a location or situation in which I don’t have access to other sources. I would suggest the authors put in the current manuscript the information that is necessary for the reader to be able to evaluate the stimuli, methods, analyses, etc. reported in the current manuscript. Readers shouldn’t have to track that information down elsewhere.

Line 226: I believe the proper spelling is “Houston”.

Lines 236-240: It takes readers more effort to understand a procedure if the order in which the tasks or events are described does not match the order in which the tasks or events were actually presented. I would suggest the authors reorganize this section to match the order in which each of the components of the experiment actually occurred.

Lines 257-260: This was confusing. The authors state the Gold-MSI involves 5 dimensions, but then they only list three: perceptual ability or engagement, musical training, and singing ability and emotion. Even if the “singing ability and emotion” are two dimensions and not 1 [in which case an Oxford comma should be used), there are still only four dimensions rather than five that are listed.

Line 263: “comprises of” sounds awkward to my ear. Perhaps “is composed of” would be better.

Line 283: What does “shallower categorization” mean? After thinking about this for a while, the best I could come up with was that this referred to the slope of transition, with categorical perception producing a steeper slope adjacent to the transition. However, the authors should explicitly describe this for readers who aren’t as familiar with categorical perception. Readers shouldn’t have to figure this out. Or if that was not what was intended, the authors should rephrase this so that such a confusion is less likely to arise.

Line 287-289: Are there data supporting the claim that people who report normal imagery do this? Or are the authors merely assuming that people who report normal imagery do this? If normal imagers don’t report doing this, why should people with aphantasia do this?

Line 313, 315: I might be missing the obvious, but what is “BKB”? I’m probably not the only reader who will be puzzled by this.

Lines 322-324: That this comparison required an auditory image is an assumption that the authors make. It seems possible (at least in principle) that other methods or formats of representation might be used in the task. What is the evidence that an auditory image was required? To be clear, I’m not saying that an image wasn’t used. I am saying the authors need to provide evidence or arguments that an image is required.

Lines 339-341: I’m not sure it’s necessary to list all these thresholds here. It might be sufficient to note the BF values in the Results section and then give the interpretation of the specific BF values that were obtained.

Lines 347-349: How were the data of participants who self-reported aphantasia used “to” the control group? I think the authors meant something like “were compared to”, but that isn’t clear here.

Line 353: A comma should be inserted before “which”.

Line 440, 450, 481, etc.: The spacing in “aphantasic/ control” is inconsistent, and there should either be a space before and after the slash (as in “lyric imagery / perceptual tone”) or a space neither before nor after the slash (in which case, change “lyric imagery / perceptual tone” to match).

Line 460: Usually a one-sentence paragraph isn’t considered to be good style. I would consider removing the paragraph break between lines 463-464.

Lines 472-474: Was this confusion admitted by the participant (perhaps during debriefing) or was this confusion determined later by the author?

Line 479: For consistency, there should probably be a hyphen in “Box Cox” (or else remove the hyphen used on line 438).

Lines 537, 550, 596, etc.: When “et al.” is used, there typically isn’t a comma used after the first author’s name.

Lines 572-573: This should be phrased more cautiously. The authors’ phrasing suggest they already believe such subgroups exist. It would be better to say something more like “to determine if subgroups exist…”

Lines 588-591: There are a couple of chapters in Hubbard’s (2018) volume Spatial Biases in Perception and Cognition that discusses and updates the idea of spatial representation of pitch.

Line 608: “may be down to the specific task used” seems a bit too colloquial.

*** end of review

6. PLOS authors have the option to publish the peer review history of their article (what does this mean?). If published, this will include your full peer review and any attached files.

Reviewer #1: No

Reviewer #2: **Yes: **Timothy L Hubbard

---

## [Author Response · Author response to Decision Letter 0]

27 Sep 2023

We thank the reviewers for their thoughtful and detailed reviews of our manuscript. We have made extensive changes to address these very helpful comments and we believe that the manuscript is now substantially improved. We have highlighted the changes that have been made in the manuscript and provide a response to each comment below. 

Response to reviewer comments

Reviewer #1: In this interesting and well executed study, the authors investigate whether participants who would qualify as aphantasic – an inability to generate mental imagery - based on a questionnaire measuring their visual imagery vividness (the VVIQ) also show reduced auditory imagery on both questionnaire measurements and objective task performance. To this end, 29 aphantasic (VVIQ < 25) and 30 control (VVIQ > 35) participants filled in questionnaires regarding their auditory imagery and musical sophistication and performed a musical imagery and a voice imagery task. The experiments and analyses are well-executed and controlled, the study investigates an interesting question that will increase our understanding of the cognitive mechanisms underlying aphantasia and caveats and alternative hypotheses are acknowledged. I only have two major comment that needs to be addressed.

Firstly, I want to applaud the authors for sharing the data and code as R markdown files that were available during review. This is a great example of open science!

We thank the reviewer for their assessment of our work. We are pleased that the reviewer thinks that the study is interesting and will increase understanding of the mechanisms underlying aphantasia. We are pleased to be able to offer the data and code for inspection during the review process. We have included further analyses and code prompted by the review – this is available via the OSF link in the manuscript.

The voice imagery task is interesting and well designed; however, I am not entirely sure to what extent it requires auditory imagery. The authors argue ‘as the CV syllables had not been heard by participants before, the participants needed to compare ambiguous voices … to an internal auditory image of the imagined vocal identities of the speaker’. However, this can be seen as a straightforward perceptual categorization task with new instances – the fact that these instances cannot be remembered but need to be compared against a category does not necessarily mean that imagery is needed for this process, right? Would the authors suggest that imagery is needed for all perceptual categorization of new instances? This needs to be discussed in more detail either in the methods section or in the discussion section as a possible pitfall of the study.

Thank you for this critique. We chose two tasks in this study. One, a version of a traditional imagery task (the pitch task) that has been frequently used to measure voluntary auditory imagery (see Aleman, Nieuwenstein, Böcker, De Haan, 2000; Gelding, Thompson & Johnson, 2015; Halpern, 1989; Janata & Paroo, 2006; Keller, Cowan, & Saults, 1995; Weir et al., 2015). 

The other task was a novel adaptation of a categorical perception task that we used to measure voice identity representations (the voice task). We chose this task and format for the following reasons:

(1) Imagery of voices is an example of auditory imagery that many people are familiar with and experience regularly. For example, people often generate auditory imagery of someone’s voice when recalling something that has been said and it’s possible to recall a sense of a famous or familiar voice in one’s ‘inner ear’. 

2) The voice task does not explicitly require participants to engage in voluntary imagery. We hoped that this would allow us to investigate imagery in a way that reduced ‘demand characteristics’. As noted by reviewer 2, one criticism of previous work (and our pitch task) is that voluntary imagery tasks may be subject to bias. Given that participants have been selected on the basis of their self-reported reduced imagery ability, they might expect to perform worse on tasks that explicitly require imagery, which might negatively influence their performance. Alternatively, as they might have an expectation of performing poorly on the task, they might be more motivated and/or attend better to the task, which might compensate for their difficulties.

3) We hypothesised that to perform well on the task, it would be beneficial for participants to generate an internal auditory image of what the words might sound like when spoken by the two voices, to categorise the vocal instances. Participants learned different vocal identities and were then presented with novel instances of those voices. These novel instances were the result of morphing between the same words spoken by the speakers to generate a continuum of sounds that transitioned between vocal identities. They were asked to categorise the voices from this continuum into the learned vocal identities. Both the ‘to-be-categorised voices’ and the specific acoustic-phonetic content of the speech had not been heard before. Hence, this was a generative task, which prevented participants from matching the heard speech to a memorised echoic template. We assumed that participants would need to generate an abstract representation of the speakers’ voice to complete the task and that participants that generated more vivid internal representations would be better able to categorise the utterances (e.g. they would have sharper categorisation functions). 

It is an empirical question as to whether our assumptions were correct. We now directly address this critique by testing for the degree of association between self-reported imagery ability and performance on the pitch and voice tasks in typical imagers. This showed that there was a correlation between performance on the pitch imagery task and self-reported auditory imagery ability. The following has been added to Page 20:

“To further understand the implications of these findings, we further correlated the auditory and visual imagery scores with accuracy on the pitch task in the typical imagery group. We used a one-tailed test as we assumed that the relationship would be positive, e.g. increased self-reported imagery would be associated with greater accuracy on the pitch task, and we had no basis to hypothesize a negative relationship. This indicated that there was a weak correlation between the auditory imagery scores (BAIS-V) and the pitch task (r = 0.32, p = 0.049) and a marginal relationship between the visual imagery scores (VVIQ) and the pitch task (rs = 0.31, p = 0.052). We further generated a composite imagery score by z-scoring the auditory and visual imagery scores and averaging them. This imagery composite score also correlated weakly with performance on the pitch task (r = 0.33, p = 0.045). From this we concluded that our pitch task indexed voluntary imagery ability, but did not distinguish between participants with aphantasia and those with typical imagery.”

In contrast, there was no relationship between self-reported auditory imagery ability and the voice task. Further, there was a significant difference in the strength of these relationships, indicating that the voice task was a poorer task for indexing imagery (as measured by self-report). As a result, we have changed references to the task from ‘voice imagery task’ to ‘voice task’ in the manuscript. We have added the additional analyses as outlined above, to the results section Page 23:

“To provide further context to this finding, we correlated the slope parameter values averaged across continua separately with the visual (VVIQ) and auditory imagery (BAIS-V) scores in turn. There was no evidence of a relationship between auditory or visual imagery self-report and performance on the voice task alone or when the measures were combined in a composite score (all ps > 0.575). We then directly tested for a difference in the strength of correlation between the auditory imagery values and the pitch and voice tasks respectively. This showed that the pitch imagery task was more strongly correlated with auditory imagery scores (BAISV) than the voice task was (z = 2.20, p = 0.028).“

We have also added additional information about the motivation for the task in the methods section (of the voice task) on Page 13:

“In this way, we hoped that this task would allow us to index auditory imagery abilities without explicitly instructing participants to generate imagery to avoid ‘demand characteristics’ associated with asking participants identified on the basis of reduced imagery to engage in an explicit imagery task.”

In addition, we have added two new paragraphs to the discussion section with regards to the task selection on Page 29 and 30.

4) Understanding whether individuals with aphantasia have difficulty with voice recognition is important in its own right irrespective of whether imagery can be demonstrated to be necessary to perform this particular task. The relationship between visual imagery and facial recognition is an area of active research. Individuals with prosopagnosia report weaker visual imagery (Gruter et al., 2009). Consistent with this, people with aphantasia report deficits in facial recognition (Milton et al., 2021; Zeman et al., 2020; Dance et al., 2023). However, it’s unclear whether this is a consequence of poor imagery (or whether they are separate deficits) and whether this reflects a specific problem with faces or a general visual recognition issue (Monzel et al., 2023). Weaknesses in perception are plausible given theoretical accounts of imagery that suggest that imagery works as ‘perception in reverse’ (cf. Pearson, 2019) and demonstration that imagery can facilitate and interfere with perception (e.g. Craver-Lemley & Arterberry, 2001). Indeed, aphantasic deficits may arise at the level of the use of prior knowledge to generate imagery and/or in sensory reactivity in response to top-down signals. Therefore, we argue that drawing a strict separation between perception and imagery may be problematic when investigating imagery deficits. 

Voice recognition is often considered to be an equivalent auditory task to facial recognition in the visual domain (Belin et al., 2011). Our voice recognition task extends previous investigations into facial recognition in aphantasia into the auditory domain. We have added some additional discussion to this effect on Page 12:

“Individuals with aphantasia report reduced facial recognition abilities (Milton et al., 2021; Zeman et al., 2020; Dance et al., 2023). The extent to which these facial recognition deficits are a consequence of poor imagery and their specificity to face rather than visual recognition more broadly is the subject of continuing research (Monzel et al., 2023). More broadly, the role of imagery in facial recognition remains unclear (Gruter et al., 2009) and specifically whether being able to see someone’s face in your mind’s eye makes you better at recognising them (Dance et al., 2023). Given that voices have been argued to function as “auditory faces” (Belin et al., 2011), we developed an equivalent voice recognition task. We hoped that this novel task would provide a complement to a more well established voluntary musical pitch imagery task.” 

Is it possible that response biases play a role in the results such that self-identifying aphants just have a higher criterion for saying that their imagery is vivid compared to others, but in fact experience the same thing? This would explain the cross-modality correlation in questionnaire scores as well as the absence of objective behavioural effects. The authors suggest the aphantasia group might not have conscious imagery and that could explain the discrepancy, but this would be a slightly different mechanism; i.e. they might experience it the same way, but are less likely to say it is vivid.

Yes – this is also an interesting possibility. We have added some content into the discussion section to introduce this possibility Page 28:

“Instead, it may be that differences only arise on self-report because the two groups differ in their thresholds for labelling an image as ‘vivid’, in that participants with aphantasia have a much higher threshold for reporting an image as vivid compared to those with typical imagery (e.g. see Dijkstra, Kok & Flemming, 2022) with these thresholds potentially also differing across modalities.”

We note that this mechanism might account for the failure to find differences in this study but not for other studies in which objective differences on imagery and imagery related tasks have been demonstrated, e.g. Bainbridge et al., (2020) and Keogh & Pearson (2018).

Minor comments:

Was there a correlation (across groups) between questionnaire measures of imagery and task performance? If not, what does this mean about the validity of these tasks for testing imagery?

See the comments further above. Thank you for this suggestion this has helped to sharpen our inferences from the study. We have included the statistics in our response above, however, the correlation between the BAIS-V and pitch task can be viewed on Page 20 and the correlation between the BAIS-V and voice task can be viewed on Page 23. In summary, our results show that our pitch task indexed voluntary imagery ability, but did not distinguish between participants with aphantasia and those with typical imagery. There was no evidence of a relationship between BAIS-V scores and performance on the voice task. Our pitch imagery task was more strongly correlated with auditory imagery scores (BAISV) than the voice task was.

N = 29 is way too small to say anything about proportions that generalize to the population.

Yes – we agree with this. We are assuming this relates to some of the content in the discussion. We have softened these statements by removing this phrasing throughout the discussion. 

Line 243, I am not entirely sure I understand the perceptual control task; was new sound created for this task? Why was not just the original song used as control?

Yes - pure tones were synthesised to match the pitch of the target lyrics of the songs presented in the lyric imagery condition. They then had to judge whether the second target tone was higher or lower in frequency. This was a purely perceptual task aimed to establish whether pitch processing abilities were similar between groups. This was included to help us interpret the results in the instance that there was a difference between the groups on the pitch imagery task. If there was a difference between the groups specifically in the imagery task (e.g. an interaction between task and group) this would indicate that the aphantasic participants had a specific difficulty in pitch imagery rather than a more general one in pitch perception. 

We used pure tones to make the task easier. The pure tones did not include vocals or additional percussion allowing participants to focus on a simple pitch decision without additional distractions and so is likely to be a purer pitch identification/discrimination task. We have explained this design choice on Page 11: 

“We used pure tones to allow participants to focus on a simple pitch decision without additional distractions that might be generated by vocals and other musical features.” 

Line 294, what are ‘CV syllables’?

This refers to Consonant Vowel syllables (e.g. Ba, Va, Ka etc). We have made this explicit in the manuscript for those unfamiliar with the term, this reads as follows on Page 13:

“In brief, pairs of the same Consonant Vowel (CV) syllables, e.g. /ka/ from different speakers were morphed.”

Was there a correlation between the control and vividness subscales of the auditory tasks and between the control of auditory imagery and vividness of visual imagery?

Thank you for this query and a related query from the other reviewer. We ran the correlation and found that it was exceptionally high (r = 0.95). For context, the correlations between these scales as reported in other literature range from r = 0.50 (Halpern, 2015) to r= 0.81 (reported by Hinwar & Lambert, 2021). This led us to review the version of the BAIS that the participants saw. While participants were told (verbally and in the written instructions) that the two parts of the BAIS (BAISV and BAISC) were two separate scales, we think because of a formatting error, participants used the BAIS-V scale to answer both sections. 

We have now acknowledged this error in the manuscript on Page 9: 

“In the current study, we only report the BAIS-V due to a formatting error in the BAIS-C Likert scale that was presented to participants.”

This is unfortunate as the BAIS-V and BAIS-C provide interesting complementary information concerning the imagery abilities of individuals with aphantasia. However, it does not affect the substantive analyses or conclusions of the manuscript. The correct scale was used for the BAIS-V (but not the BAIS-C) which was the most important of the two measures and provides the most appropriate, analogous auditory scale to the visual imagery scale (VVIQ).

References

 Halpern, A. R. (2015). Differences in auditory imagery self-report predict neural and behavioral outcomes. Psychomusicology: Music, Mind, and Brain, 25(1), 37–47. https://doi.org/10.1037/pmu0000081

Hinwar, R. P., & Lambert, A. J. (2021). Anauralia: The Silent Mind and Its Association With Aphantasia. Front Psychol, 12, 744213. doi:10.3389/fpsyg.2021.744213

Reviewer #2: *** beginning of review

General Issues:

A. Epiphenomenon. The authors offer speculation why there might not be a difference in performance on auditory imagery tasks between participants who self-identified as having aphantasia and matched control participants (e.g., lines 101-109), but one explanation that the authors did not discuss is that imagery is not causal in the imagery tasks but is simply an epiphenomenon of the mechanisms that are causal. An analogy from study of visual imagery (which I think is due to Pylyshyn, but I’m not sure of the source) involves the light on a desktop computer that begins blinking when the computer is booting up. That blinking light plays no causal role in the functioning the computer, and the computer could complete booting-up just as well if that light was burned out. In the case of imagery, images are like that blinking light in that they are just a byproduct of the booting up process and play no causal role in that process. I personally am not convinced by previous arguments in the literature that imagery is epiphenomenal and does not play a causal role in cognition, but the author’s data can be interpreted as supporting that position. This epiphenomenal alternative should be discussed in the manuscript.

Thank you for this suggestion. We have added some sentences to this section to refer to what current findings in aphantasia literature might mean for epiphenomenal accounts of imagery. We have included some discussion of this throughout the manuscript. To the introduction Page 4, we have added the following:

“… it is not always the case that overt differences in imagery behaviour have been found when comparing people with aphantasia to typical imagers. This might suggest a disconnect between subjective experience of imagery and imagery behaviour and more broadly, might align with theoretical accounts suggesting that imagery phenomenology does not play a functional role in cognitive processes (Pylyshyn, 2003). Indeed, when individuals with aphantasia have been presented with batteries of tasks thought to involve imagery, differences in behaviour have sometimes been shown to be minimal. Specifically, differences were apparent only in a subgroup of participants that report the lowest imagery, were found in only one or a small number of imagery tasks in the battery, were apparent in response time (but not accuracy) or were shown only at the highest levels of difficulty (Keogh & Pearson, 2021; Milton et al., 2021; Monzel, Keidel & Reuter, 2021; Pounder et al., 2021; Jacobs et al., 2018; Liu & Bartolomeo, 2023). However, whilst subtle, these differences might equally indicate qualitatively different or weaker imagery, which in turn might be interpreted as evidence against epiphenomenal accounts.”

To the discussion on Page 28, we have added:

“…Our findings might support the broader assertion that the experiential phenomenology of mental imagery is epiphenomenal and does not play a causal or functional role in cognition (Pylyshyn, 2003). However, we note that these explanations cannot account for previous findings showing positive differences on behavioural imagery and imagery related tasks in people with aphantasia (Bainbridge et al., 2020; Keogh & Pearson, 2018)“ 

B.Thresholds. another alternative that should be addressed is the possibility that the participants who self-reported aphantasia and the matched control participants might have equally vivid levels of imagery, but the participants who self-reported aphantasia have a much higher threshold for labeling an image as vivid. In other words, an image of a given level of vividness might be labeled as “strong” or “vivid” by person who does not self-identify as having aphantasia but as “weak” and “not vivid” by a person who does self-identify as having aphantasia. This possibility occurred to me at a couple of different locations in the manuscript (e.g., lines 364-383, 412-414, 427-429, 521-543) and should also be discussed in the manuscript.

Thank you for this comment which was also mentioned by the other reviewer, we have now included some discussion of this possibility on Page 28:

“Instead, it may be that differences only arise on self-report because the two groups differ in their thresholds for labelling an image as ‘vivid’, in that individuals with aphantasia have a much higher threshold for reporting an image as vivid compared to those with typical imagery (e.g. see Dijkstra & Fleming, 2023) with these thresholds potentially also differing across modalities.”

C. Task Administration. The authors provide reasonable descriptions of the different experimental tasks. However, information regarding the experimental procedures and the administration of the experimental tasks is missing. In what order were the tasks given? Was order counterbalanced or randomized, and if not, then why not? How long did the completion of all the tasks require? Was this all done in a single session for each participant? Were participants always run one at a time, or were they run in groups? Etc.

The study was undertaken in one single testing session, lasting approximately 90 minutes. Each participant was tested one by one. Participants first undertook the VVIQ, Gold-MSI and BAIS, followed by three tasks (the voice imagery task, the musical imagery pitch task and a visual-spatial statement task that is not detailed in the manuscript, that were presented in a randomised order. The imagery and perceptual conditions of the musical imagery pitch task were ran consecutively, but the order of this was counterbalanced across participants. We have added a Procedure section to the manuscript that contains this information on Page 15.

D. Demand characteristics. Through its history, imagery research has always struggled with the issue of demand characteristics (i.e., the possibility that participants can deduce the experimental hypothesis and then modify their responses to reflect what they think the experimenter wants to observe). Given that participants who self-reported aphantasia could presumably deduce the authors’ interests in aphantasia (after all, those participants were recruited from aphantasia websites), what precautions were taken to limit any possible role of demand characteristics or to assess whether demand characteristics contributed to the data? This needs to be discussed.

We agree that this is an issue in much of the literature. This was partly the motivation for the design of the voice task, we hypothesised that it would be beneficial to performance for participants to generate an auditory image of the speakers voices but we did not explicitly ask them to generate an auditory image. It is indeed possible that participants might have guessed the purpose of the pitch task. This could have two outcomes. Aphantasic participants might have anticipated that they would be worse at this task given their self-reported weaknesses in imagery which might have reduced their performance on the task. This seems unlikely given that performance was similar between the groups. Another alternative is that the aphantasic participants would assume that they would be worse at the task which might act as an additional motivation and they might concentrate more – such that this increased their performance – and allowed them to perform at a similar level due to the application of increased attention and effort, which may have compensated for their imagery deficits. We have added the following two paragraphs to the discussion of the possible effects of demand characteristics on Page 29:

“We chose two tasks. One was a traditional voluntary pitch-based imagery task; performance on this task was correlated with self-reported auditory imagery vividness but it did not identify a difference in performance between groups. Given that this task explicitly asked aphantasic participants to generate auditory imagery, it may have been subject to ‘demand characteristics’. However, one might assume that such a bias would lead to reduced performance in the aphantasic group. Given that performance was equivalent between the groups, it is possible that aphantasic participants were more greatly motivated to perform well on this task given their self-reported difficulties, which acted to compensate for their imagery deficits. However, this would make the strong assumption that imagery ability in aphantasia can be upregulated by attention or similar factors.

Our voice task was less susceptible to ‘demand characteristics’. However, we did not find evidence that performance on this task was correlated with auditory imagery self-report and as such it may not have been an effective task for measuring auditory imagery ability. Despite this, the absence of differences between the groups on this task is interesting, especially given recent findings of facial recognition deficits in individuals with aphantasia (Milton et al., 2021; Zeman et al., 2020), albeit these deficits may have been driven by a more general visual deficit (Monzel et al. 2021, but also see Pounder et. 2022). Here, in the auditory domain we did not find evidence of an equivalent deficit for voice recognition. Further research is required examining the relationship between perception and imagery across a range of tasks and modalities in individuals with aphantasia.”

E. Continua. I’m not clear regarding the nature of the continua that underlay the morphing of speaker (frequency? timbre? voice onset time? just a mix of “whatever”? etc.) There are numerous possible dimensions along which speech stimuli (or auditory stimuli more broadly) can differ, and if the effect of morphing is going to be related to imagery, readers need to know what underlying dimensions of the stimuli are involved. This becomes more important given that there has been some debate regarding which properties of auditory imagery are necessary and which are optional (e.g., Intons-Peterson has previously argued that loudness is not a necessary property of auditory imagery). Also, there might have been different features changed in different continua, and if so, then the dimensions along which change occurred might be confounded. This needs to be discussed.

We used STRAIGHT to synthesise the speech. This software is frequently used in studies of categorical perception of speech (see Rodgers & Davis, 2009; Davis et al., 2019; Rodgers & Davis, 2017). This kind of morphing changes multiple acoustic parameters simultaneously via interpolation. The advantage of this kind of morphing is that the resulting speech is very natural sounding and voice identity is preserved unlike synthetic continua which does not sound like an authentic human voice. There is no straightforward way to quantify which acoustic features have been changed and how this affected the outcome of the study. We used multiple continua to deliberately introduce some variance over the acoustic/phonetic features over which vocal identity was communicated. We have added some more information in the method section to explain this on Page 14:

“This morphing approach uses interpolation to create intermediate sound tokens between the two end point stimuli. The percept of change of speaker that is expressed across the continua reflects changes across multiple simultaneously changing acoustic features. This approach provides highly natural sounding speech tokens but does not make it possible to identify how individual acoustic features are influencing speaker identify judgements. We used a number of continua to ensure that multiple acoustic features would contribute to voice identification.” 

F. Parallel Construction. There are a few places in the manuscript in which the authors need to use parallel construction. For example, the authors refer to “impairments” in line 61 but “experiences” in line 63, but I think they also meant “impairments” in line 63. In another example, they refer to “individuals” in page 515 and “participants” in line 516, but they should use the same term in both places. In general, using multiple terms for the same idea, construct, or entity makes more work for the reader, and so terminology should be more consistent across the manuscript. I’ve pointed out two instances, but the authors should check the document for other instances. On a related note, there are inconsistencies in the way that references within parenthetical reference lists are organized. The organization sometimes appears alphabetical by first author and not chronological (lines 58-59; 68-69), chronological but not alphabetical by author (e.g., line 89) or neither chronological nor alphabetical (e.g., lines 97-98). This should be more consistent throughout the manuscript.

We have tried to address these parallel constructions. We use the term ‘participants’ to refer to the people in the study and in reference to those that took part in other experiments. We refer to ‘individuals’ when talking about people with aphantasia in general (or as a result of findings from these studies). We use ‘impairment’ only in reference to a deficit or difficulty and ‘experience’ to a more general term to describe what a person may be experiencing (that is not necessarily associated with an impairment). 

G. Figures. Figures 1, 2, and 4 are very faint in my printed copy of the manuscript. I would suggest using a higher contrast and more saturation. There does not appear to be a Figure 3 in my copy of the manuscript, and I don’t know if this is a simple labeling error or if an intended figure was actually omitted.

Yes – please accept our apologies this was a labelling error – there are only three figures. One of the figures was appended as an image in a multi-panel figure but the figure legend had not been updated to reflect this. This is now corrected on Page 18.

The figures had appropriate contrast and saturation when uploaded. If the article is accepted for publication, we will make sure that the figures are reproduced correctly at the proofs stage and will make any changes as necessary.

H. Paragraph lengths. There are several paragraphs that are too long (.e.g., lines 85-113, 114-146, 544-573, 574-601). Breaking these into 2-3 smaller paragraphs (and adjusting the organization as needed) would make this easier on the reader.

Thank you for this style suggestion. We have shortened these paragraphs.

I. References. There are a number of problems with the References. Pounder et al. (2021) is not in the References section, although there is a Pounder et al. (2022) that might have been meant. Similarly, Banno et al. is listed as (2006) in the main text but (2007) in the References section. Crowder et al. (1989), Jacobs et al. (2018), Keogh and Pearson (2021), Knauff et al. (2000) are cited in the main text but are not listed in the References section. Furthermore, Alderson-Day and Fernyhough (2015), Berry and Laskey (2012), Bench et al. (1979), Brogaard and Gatzia (2017), Emerson and Laskey (2012), Fegen et al. (2015), Floridou et al. (2015), Goggin et al. (1991), Holmes et al. (2004), Keogh et al. (2021), Lavan et al. (2019), Perrachione et al. (2011), Perrachione et al. (2009), Pounder et al. (2022), Price (2012), Scott and McGettigan (2015), Shergill et al. (2001), Vilhauer (2016), and Williamson et al. (2012) are all listed in the References section but are not cited in the main text. There are problems with individual citations as well (e.g., formatting of Box and Cox, 1994; Crowder, 1989; spelling in Dawes et al., 2022; Lui and Bartolomeo is listed as “2023” in the text but as “under review” in the text). Zatorre and Halpern (1993) is listed twice. I am sorry of this sounds overly harsh, but I think the authors need to know that when any reviewer sees this many problems with the References involving issues that should have been checked and corrected before submission, it implies to that reviewer that the authors haven’t done due diligence and were not sufficiently careful in preparing the manuscript (i.e., if the authors appeared this careless with their references, a reviewer would likely question whether the authors were sufficiently careful in preparing the rest of the manuscript [including the details of method, results, and analyses]).

We are sorry about problems with the references. This manuscript has been worked on by multiple authors and additional references have been added and removed multiple times during the editing and writing process. We have now double checked the references and we hope that they are in order.

Specific Issues:

Line 44: I would delete the comma after “controls”. More importantly, the phrasing suggests the same individual had to self-identify as both having aphantasia and as a matched control, which obviously isn’t what the authors intended,

Thank you for this suggestion it has been acted on and we have removed the comma from this sentence, which now reads (in the Abstract on Page 2):

“In the current study, individuals that self-identified as being aphantasic and matched controls, performed two tasks requiring auditory imagery: a musical pitch-based imagery and voice-based categorisation imagery task”

Lines 69-71: Why is this claimed to be heterogeneous? If the same patterns are occurring across modalities, wouldn’t that suggest a more homogenous (or similar) mechanism? Am I missing something? Also, what does it mean to report “associations” in imagery deficits? Are the authors referring to something like a difficulty in imaging visual height is associated with a difficulty in imaging auditory pitch (cf. lines 586-595)? It seems like it would be easier to just say something like “report related deficits in multiple modalities.”

We think that heterogeneous is the correct expression, but we need to make our meaning clearer in this section. The point we are making is that some people have selective deficits, e.g. they only report a deficit in one modality, and others have deficits across multiple modalities. Therefore, the pattern of deficits across modalities seems highly specific to each individual and may vary across individuals. We hope that we have made our meaning clearer now in this paragraph, on Page 5:

“The heterogeneity in the profile of imagery abilities in those with aphantasia (e.g. Zeman et al., 2020) and in those with typical imagery more broadly (McKelvie, 1995), raises important questions about the necessity of imagery.”

Line 111: “the experience of people with aphantasia...” Isn’t the point that people who self-report having aphantasia don’t have an experience (by definition)? Or are the authors referring to a potential non-imaginal experience?

We thank the reviewer for this comment, we have changed this word to ‘capabilities’ to avoid any confusion. On Page 5, this sentence reads:

“Further research, in a broader range of tasks, is required to build a better picture of the capabilities of people with aphantasia, especially in understanding whether self-reported deficits translate to observable differences in behaviour in imagery domains beyond vision. “ 

Lines 134-136: Can “primary sensory… respectively” be stated more simply? It won’t be clear for many readers.

We hope this sentence is clearer now, this sentence reads on Page 6: 

“For example, primary sensory and secondary association cortex are jointly activated by perception and imagery in the temporal and occipital cortices respectively for audition and vision.”

Line 147: The antecedent of “other” is not clear. Also, “people with aphantasia” would seem to include all imagery modalities, so what would be “other” than all imagery modalities? The authors probably meant something like “multiple” or maybe “visual”, but that isn’t what they actually say.

Thank you for this suggestion. “Other” has been changed to “multiple”, and now reads on Page 7:

“In summary, people with aphantasia often self-report deficits in multiple imagery modalities, including in the auditory domain.”

Lines 151-154: This phrasing implies the authors are expecting the matched control group to also report deficits. Also, the phrasing implies the participants tested themselves.

We have rephrased this section and have specified that this in relation to aphantasia. This sentence reads (on Page 7):

“We assessed how many of the participants with aphantasia, recruited on the basis of their reduced visual imagery, also reported concomitant auditory imagery deficits.”

Lines 157, 549: Do the authors really want to refer to aphantasia as a comorbidity or a morbidity, especially in light of the fact that aphantasia doesn’t seem to hinder normal functioning?

Yes – we agree - the reference to co-morbidity has been removed.

Lines 191, 215, 219, 279: This is a misuse of the term “paradigm”, although I admit it is a misuse that appears common in the psychological literature. A “paradigm” refers to the axioms, methods of investigation, and appropriate topics of investigation underlying a broad area of study. Examples of paradigms within psychology include behaviorism, information processing, and connectionism (neural networks). The term “paradigm” does not apply to the methodology or design of a single experiment or small set of experiments.

The word ‘paradigm’ has been changed throughout to ‘task’ to reflect this critique.

Line 205: Is this correct? Why does an anchor term for the control subscale refer to the vividness of the imagery rather than to the ease with which the initial image could be transformed into another image (i.e., controlled)? If this is correct, it would seem to offer a significant challenge to the validity of the BAIS-C subscale.

Many thanks for raising this descriptive error. The first reviewer enquired with regards to the correlation between the BAIS-V and BAIS-C subscales, which was exceptionally high r = 0.95. We reviewed the version of the BAIS that the participant were given. This showed that the same Likert scale was used for both the BAIS-V and BAIS-C due to a formatting error. 

As per our previous comment to reviewer 1, we have now acknowledged this error in the manuscript (on Page 9). This is unfortunate as the BAIS-V and BAIS-C provide interesting complementary information concerning the imagery abilities of individuals with aphantasia. 

The correct scale was used for the BAIS-V (but not the BAIS-C). This vividness scale is the better of the two scales for providing a complementary auditory vividness scale to the equivalent visual imagery scale (VVIQ). Hence, this error does not affect the substantive analyses or conclusions of the manuscript. 

Lines 224, 293, etc.: As a reader, I never like it when I am directed to another source to find methodological and other information that is important for evaluating the article I am reading. I might be reading in a location or situation in which I don’t have access to other sources. I would suggest the authors put in the current manuscript the information that is necessary for the reader to be able to evaluate the stimuli, methods, analyses, etc. reported in the current manuscript. Readers shouldn’t have to track that information down elsewhere.

Thank you for this comment. We have added some additional text into the main manuscript and do not require supplementary materials any longer.

Line 226: I believe the proper spelling is “Houston”Thank you for spotting this spelling mistake. This has been corrected on Page 10.

Lines 236-240: It takes readers more effort to understand a procedure if the order in which the tasks or events are described does not match the order in which the tasks or events were actually presented. I would suggest the authors reorganize this section to match the order in which each of the components of the experiment actually occurred.

We have reorganised the materials section so that the questionnaires are listed first, followed by the details of the two tasks in the methods section.

Lines 257-260: This was confusing. The authors state the Gold-MSI involves 5 dimensions, but then they only list three: perceptual ability or engagement, musical training, and singing ability and emotion. Even if the “singing ability and emotion” are two dimensions and not 1 [in which case an Oxford comma should be used), there are still only four dimensions rather than five that are listed.

This reflected the positioning of an errant comma. These are the 5 stand-alone dimensions: (1) self-reported perceptual ability, (2) active musical engagement, (3) musical training, (4) self-reported singing ability and (5) sophisticated emotional engagement. The manuscript has been updated to reflect this on Page 9 and Page 10.

Line 263: “comprises of” sounds awkward to my ear. Perhaps “is composed of” would be better.

This has been changed to your suggested wording.

Line 283: What does “shallower categorization” mean? After thinking about this for a while, the best I could come up with was that this referred to the slope of transition, with categorical perception producing a steeper slope adjacent to the transition. However, the authors should explicitly describe this for readers who aren’t as familiar with categorical perception. Readers shouldn’t have to figure this out. Or if that was not what was intended, the authors should rephrase this so that such a confusion is less likely to arise.

Apologies that this was not clear. Shallower categorisation means a less consistent response. This translates as more uncertainty as to which of the two vocal identities the participant assigned the sounds to. We have added some detail to make this clearer to on Page 13.

“Shallower functions would reflect greater uncertainty in labelling the vocal identities, in a similar manner to the way in which shallower identification functions are interpreted as reflecting less well specified phonological representations of speech sounds in those with language learning impairments (e.g. Melby-Lervåg et al., 2012). In this way, we hoped that this task would allow us to index auditory imagery abilities without explicitly instructing participants to generate imagery to avoid ‘demand characteristics’ associated with asking participants identified on the basis of reduced imagery to engage in an explicit imagery task.”

Line 287-289: Are there data supporting the claim that people who report normal imagery do this? Or are the authors merely assuming that people who report normal imagery do this? If normal imagers don’t report doing this, why should people with aphantasia do this?

This is an assumption in our task, and a well established assumption within the literature for categorical perception tasks. Numerous studies e.g. Bomert & Mitterer (2004); Melby-Lervåg, Lyster, Hulme (2012), suggest that shallow categorical perception functions are suggestive of “less vivid” internal representations. However, there is no explicit indication of the format of these representations (despite the term ‘vivid’ having very visual connotations). We have now removed reference to the format of the internal image, and this reads on Page 13:

 “To categorise a novel ambiguous vocal stimulus as belonging to a particular vocal identity, one draw upon an internal representation of the to-be-categorised vocal identities to compare the novel auditory stimulus against.”

We have also analysed the relationship between self reported auditory imagery (BAIS-V) and accuracy of the two tasks. The pitch but not the voice task correlates with auditory imagery report (detailed on Page 20 and 23 of the manuscript). We have included these analyses in the manuscript. Given this finding we now discuss the extent to which auditory imagery was necessarily required to perform the voice task – see Page 30.

Line 313, 315: I might be missing the obvious, but what is “BKB”? I’m probably not the only reader who will be puzzled by this.

This refers to the Bamford-Kowal-Bench sentence set, and we have added the reference for this set (Bench, Kowal, & Bamford, 1979) on Page 14.

Lines 322-324: That this comparison required an auditory image is an assumption that the authors make. It seems possible (at least in principle) that other methods or formats of representation might be used in the task. What is the evidence that an auditory image was required? To be clear, I’m not saying that an image wasn’t used. I am saying the authors need to provide evidence or arguments that an image is required.

This was an assumption. As you will see from our previous responses, we have now addressed this empirically on Page 20 and Page 23 and have added several paragraphs to the discussion on Page 29-30 with regards to this point.

Lines 339-341: I’m not sure it’s necessary to list all these thresholds here. It might be sufficient to note the BF values in the Results section and then give the interpretation of the specific BF values that were obtained.

We thank you for this comment, however, we think it important to include an overview of these thresholds to inform readers (who may be unfamiliar with BF values) of how to interpret these values. This is a typical practice in articles that use Bayesian analyses. For example, this is how we have reported BF in our previous manuscript (Pounder et al., 2022) and how other research groups have also reported this information in their results section (Dawes, Keogh, Andrillon & Pearson, 2020). Given that it doesn’t take up many words, we would prefer to keep it in.

Lines 347-349: How were the data of participants who self-reported aphantasia used “to” the control group? I think the authors meant something like “were compared to”, but that isn’t clear here.

This has been changed to ‘compared to’.

Line 353: A comma should be inserted before “which”.

This comma has been added.

Line 440, 450, 481, etc.: The spacing in “aphantasic/ control” is inconsistent, and there should either be a space before and after the slash (as in “lyric imagery / perceptual tone”) or a space neither before nor after the slash (in which case, change “lyric imagery / perceptual tone” to match).

This has been changed throughout. An additional space has been added after “aphantasic”, and an additional space after “lyric imagery”.

Line 460: Usually a one-sentence paragraph isn’t considered to be good style. I would consider removing the paragraph break between lines 463-464.

This has been changed and we have removed this paragraph break.

Lines 472-474: Was this confusion admitted by the participant (perhaps during debriefing) or was this confusion determined later by the author?

It was clear from the data that they systematically mislabelled the voices, e.g assigned speaker 1 to speaker 2 and vice versa. We have added the following to the manuscript on Page 23:

“It was clear from the data that they systematically mislabelled the voices, e.g assigned speaker 1 to speaker 2 and vice versa.” 

Line 479: For consistency, there should probably be a hyphen in “Box Cox” (or else remove the hyphen used on line 438).

A hyphen has been added as requested to Box-Cox (on Page 21 and Page 23 of the manuscript)

Lines 537, 550, 596, etc.: When “et al.” is used, there typically isn’t a comma used after the first author’s name.

We have revised all references, many thanks once again for bringing this to our attention 

Lines 572-573: This should be phrased more cautiously. The authors’ phrasing suggest they already believe such subgroups exist. It would be better to say something more like “to determine if subgroups exist…”

This has been changed.

Lines 588-591: There are a couple of chapters in Hubbard’s (2018) volume Spatial Biases in Perception and Cognition that discusses and updates the idea of spatial representation of pitch.

Many thanks for highlighting this useful resource. We have added some relevant information with regards to pitch and how this may be represented by other sensory means e.g. through touch. The following has been added to Page 28-29:

“Pitch is often described using spatial terms and is organised on a scale of low to high (Connell et al., 2013) giving rise to the concept of ‘pitch height’ along a vertical location which may facilitate re-mapping of pitch to a spatial representation (Rusconi et al., 2006). For instance, pitches that are ‘high’ are perceived to be situated in higher visuospatial location (e.g. ‘up’ in space) compared to lower pitches (Carnevale & Harris, 2016; Parise et al., 2014), with a similar mapping existing between auditory pitch and tactile location (Deroy et al., 2016). 

…..Conversely, in the musical imagery pitch task, while the experimenter was present in the room to ensure participant primarily did not hum or sing, participants could have mapped this through touch, for example, made small sweeping motions (unnoticed by the experimenter) with their finger in order to aid with spatial mappings (indeed, there are wide variations amongst individuals between the sensory attributes and features that are associated with these mappings, see (Deroy et al., 2016; Spence, 2011)”

Line 608: “may be down to the specific task used” seems a bit too colloquial.

We removed/edited this paragraph, and this sentence is no longer in the manuscript.

*** end of review

---

## [Decision Letter · Decision Letter 1]

12 Oct 2023

PONE-D-23-05751R1No clear evidence of a difference between individuals who self-report an absence of auditory imagery and typical imagers on auditory imagery tasksPLOS ONE

Dear Dr. Evans,

Thank you for submitting your manuscript to PLOS ONE. After careful consideration, we feel that it has merit but does not fully meet PLOS ONE’s publication criteria as it currently stands. Therefore, we invite you to submit a revised version of the manuscript that addresses the points raised during the review process.

We look forward to receiving your revised manuscript.

Kind regards,

Jie Wang, Ph.D.

Academic Editor

PLOS ONE

**Additional Editor Comments:**

The authors should address the major and minor issues raised by the two reviewers.

Reviewers' comments:

Reviewer's Responses to Questions

**Comments to the Author**

1. If the authors have adequately addressed your comments raised in a previous round of review and you feel that this manuscript is now acceptable for publication, you may indicate that here to bypass the “Comments to the Author” section, enter your conflict of interest statement in the “Confidential to Editor” section, and submit your "Accept" recommendation.

Reviewer #1: (No Response)

Reviewer #2: (No Response)

2. Is the manuscript technically sound, and do the data support the conclusions?

Reviewer #1: Yes

Reviewer #2: Yes

3. Has the statistical analysis been performed appropriately and rigorously? 

Reviewer #1: Yes

Reviewer #2: Yes

4. Have the authors made all data underlying the findings in their manuscript fully available?

Reviewer #1: Yes

Reviewer #2: Yes

5. Is the manuscript presented in an intelligible fashion and written in standard English?

Reviewer #1: Yes

Reviewer #2: Yes

6. Review Comments to the Author

Reviewer #1: The authors fully addressed most of my previous comments.

The only point I think could due with some additional discussion is the correlations between the questionnaires and the auditory pitch task. If I understood this correctly, these two measurements are positively correlated in the control group – suggesting that subjective reports of vividness correlate with objective performance in this task – but there is no difference between the aphantasic and control groups, which are themselves defined based on the subjective reports (low VVIQ versus high VVIQ). The authors suggest that this might be because ‘aphantasic participants were more greatly motivated to perform well on this task given their self-reported difficulties, which acted to compensate for their imagery deficits.’ But this seems a little far-fetched to me. Are other explanations possible? For example, perhaps aphantasic participants use different strategies to perform the same task (e.g. as suggested by Koegh, Wicken & Pearson, 2021 for working memory). Could the authors speculate more on this?

Reviewer #2: *** beginning of review

General issues:

Relationship of Aphantasia to Perception. The authors address the possibility of aphantasia across different modalities in imagery. Even so, I found myself wondering (e.g., while reading lines 70-83) about the relationship between aphantasia in a given modality and perceptual functioning within that same modality. Given that there is a correlation between patterns of brain activation during perception in a given modality and imagery in that same modality, one suspects aphantasia might be related to deficits in perception. This would also seem related to the question of whether imagery draws on a single unitary mechanism or multiple cognitive mechanisms (cf. line 83). Although the relationship between imagery and perception is briefly addressed on lines 143-149, perhaps this might be moved to earlier in the introduction and expanded. Indeed, if auditory aphantasia is not linked to any auditory perceptual deficits, then that would have significant implications for understanding such aphantasia, as well as understanding of imagery more generally.

Specifying the Modality. There are multiple instances in which in the authors refer to “mental images”. Sometimes it is not clear if they are referring to both visual images, auditory images, or both (unimodal) visual and auditory images (e.g., lines 419, 421, 448, etc.). Relatedly, visual imagery and auditory imagery are treated as separate types of unimodal images, but would the same patterns of aphantasia occur if a single multimodal image were generated? Also, I would suggest that rather than the general “mental imagery” the authors always use the more specific “visual mental imagery” or “auditory mental imagery” whenever possible. There are also a few instances in which the authors refer to “aphantasia participants”, but it isn’t clear whether the aphantasia being referred to is visual or auditory (e.g., line 533).

“Control Group” vs. “Typical Imagery Group”. The authors seem to alternative between referring to their control group as the “control group” (e.g., lines 388, 390, 413, 458, etc.) and referring to their control group as the “typical imagery group” (e.g., lines 448, 453, 456, 476, etc.). Using multiple terms for the same construct or group makes more work for the reader, and so the authors should just pick a single term and use that term more consistently.

Specific Issues:

Lines 35-36: Line 35 states “it is possible to generate many different kinds of internal representation”, which implies generation of non-imagery forms of representation as well as generation of imagery, but all the examples listed on line 36 seem to involve imagery.

Lines 61-62: “are well documented” might be a bit strong. Perhaps “are relatively well documented” might be better.

Line 78: Perhaps “in those domains” should be inserted after “require imagery”.

Line 105: What are “frank differences” and how do they differ from just “differences”? If there is a difference, it should be explained, and if there is no difference, perhaps “frank” should be deleted. Relatedly, what is a “frank deficit” (on line 168), and how does that differ from just a “deficit”? If there is a difference, it should be explained, and if there is no difference, perhaps “frank” should be deleted.

Lines 106-110: The statement “it is not clear… one explanation could be…” seems disorganized. More specifically, it seems strange to me that the authors start the second sentence with “one explanation…” given that they just spend lines 90-105 discussing other possible explanations for why differences were not more apparent. (e.g., imagery is epiphenomenal, qualitatively different or weaker imagery, etc.). Rather than making line 106 sound like a shift to a different topic (an explanation rather than a description), it might be better to make it clearer that additional explanations will be given.

Lines 130-134: A greater number of beats in an auditory image and a longer distance in a visual image are stimulus-specific properties of the images, and it isn’t entirely clear why a common mechanism would respond to these different types of stimulus-specific properties, and so perhaps some speculation on this might be offered. Also, for consistency “ones” in line 130 should be replaced with “images”.

Line 140: For clarity, I would add “the amount of” before “grey”

Lines 173-181: Was the VVIQ score used in recruiting participants, or was it only obtained after recruitment? The text implies the former, but I suspect the latter actually occurred. Regardless, this should be clarified. Also, did the forums from which the authors recruited participants distinguish between visual or auditory aphantasia? More specifically, did the participants actually self-identify as aphantastic based on their visual imagery, or might some self-identify as aphantastic based on their auditory imagery? Also, would not have recruiting on the basis of auditory aphantasia (rather than recruiting based on visual aphantasia) have also allowed examination of the relationship between visual aphantasia and auditory aphantasia? Other than possibly having more total respondents if they advertised for visual aphantasia, why didn’t the authors recruit from the population that they were really interested in, which was auditory aphantastics?

Line 176, 217, 298: I think using the Oxford comma usually improves clarity, and so I would insert a comma after “Twitter” on line 176, after “preferences” on line 217, and after “emotion” on line 298. Of course, if I missed any lists of three or more items that didn’t already include an Oxford comma, then analogous comments would apply to those lists.

Lines 209-212: I realize the data for the control subscale will eventually be discarded, but this subscale should still be fully described. An example question from the subscale should be given and the types of transformations described. Also, anchor terms for both the vividness subscale and the control subscale should be provided.

Line 218: A space should be inserted after “behaviour”.

Lines 228-232: The authors refer to the General Musical Sophistication Scale as one part of the Goldsmiths Musical Sophistication Index, but when they describe scoring of the Scale, the claim “these values are summed to provide an overall score for the participants” sounds as if they are describing the Index (the Gold-MSI) rather than a separate General Musical Sophistication Scale. This needs to be clarified.

Line 258: For consistency with the latter part of the sentence, “underlined” should be inserted after “second”.

Line 297: A space should be inserted after “properties”.

Line 304: “can” should be inserted before “draw” or “draw” should be changed the “draws”. The former would be preferable.

Lines 304-312: But is imagery really needed for the voice task? The voice task is intended to be an analogue of facial recognition, but the authors already admit that “the role of imagery in facial recognition remains unclear” (lines 281-282). I’m not suggesting that imagery can’t be used in the voice task; it might be that at least some participants used imagery at least some of the time. What I am suggesting is that it isn’t clear that imagery is required for the voice task, either for the aphantasia group or for the control group. Perhaps voices might be characterized based on descriptions of some features rather than imagery per se.

Lines 337-338: Why were different CV syllables used for make speakers and for female speakers? In other words, why wasn’t the same set of two (or four) CV syllables used for both male speakers and female speakers?

Line 342: Maybe I just missed it, but what is a Bamford-Kowal-Bench sentence? How does it differ from other types of sentences? I appreciate that the authors added a reference, but this should still be clarified in the text.

Line 357: I’m not clear what “presented one by one in a fixed order to reduce confusion” means. Wouldn’t the pairs be presented one-at-a-time anyway? How does presenting a fixed order reduce confusion (e.g., were not the male voices clearly distinguishable from the female voices)? Could this fixed order supply or allow additional cues that might make imagery more or less necessary or useful?

Line 361: “each participant was tested one-by-one” should either be “participants were tested one-by-one” or “each participant was tested individually”. “each” implies a single participant, and you can’t have a single participant tested “one-by-one”.

Line 362: The comma after “BAIS” should be deleted (or if not, then I don’t understand the sentence).

Line 374-375: I’d insert a comma after “met” and delete the comma after “conducted”.

Line 378-381: I still don’t think the authors need to list all the BF levels here. Listing interpretations for all BF values is like mentioning that p values greater than .05 won’t be considered significant in the presentations of the Mann-Whitney U tests or ANOVAs. Giving the interpretation of the BF that was actually obtained is sufficient. I leave this up to the action editor.

Line 393: “data… was”. Unless the authors are referring to the android in Star Trek, “data” is a plural term, and so “data… were” is correct (the singular form is “datum”).

Lines 437-442: This was confusing. The authors state that the analysis was carried out using d prime values (line 438), but the results were reported in terms of accuracy (line 441). What was the dependent variable? D prime? Or accuracy?

Lines 470-471: “performance on” should be inserted before both instances of “the pitch task”, as the correlation with imagery does not involve the task per se, but rather involves performance on the task.

Line 487: Box and Cox (1964) is still not in the References section.

Lines 499-501: Whether the effect of participant group was significant or not should probably be mentioned, as well.

Line 517: If the authors wish to conform to APA style, then abbreviations such as “e.g.,” and “i.e.,” are used only within parenthetical expression. Within the main text, these should be spelled out (e.g., “for example”, “that is”, etc.)

Line 525: Although the readers question of “95th what?” is answered by the end of the sentence, just seeing “above the 95th” does seem a little odd.

Line 543: How was the combining of VVIQ and BAIS-V into a composite score done? Were the individual scores added, averaged, or was some other method of combination used?

Line 546: “BAISV” has been hyphenated previously in the manuscript (e.g., lines 393, 406, 411, 423-426, 541, etc.), and so should be hyphenated here for consistency.

Line 573: Well, the tasks might not actually require auditory imagery if the objective performance on those tasks by people who claim to not experience auditory imagery does not differ from the objective performance by people who do claim to experience auditory imagery. Just because imagery might offer one strategy to complete a task does not mean that participants necessarily use that strategy (e.g., see Zatorre & Halpern’s, 2005, and Hubbard’s, 2010, argument why simply observing a given pattern of responses when participants are instructed to use imagery does not establish that those participants did in fact use imagery; see also Hubbard’s, 2018, discussion of representational ambiguity). Claiming that the tasks required auditory imagery is probably too strong, and it would be much more prudent to weaken this.

Lines 610-611: Perhaps this can be elaborated?

Lines 631-632: But the differences are evident. Perhaps the authors intended to say that the reasons for the differences, rather than the differences per se, are not evident?

Line 639: There is an extra opening parenthesis on this line.

Lines 650-651: Plack et al. and Yuskaitis et al. are cited in the main text but are not listed in the References section.

Lines 658-660: “suggest” should be “suggests”. Also, an example of such evidence (even if parenthetical) would be helpful to readers.

Line 663: What is “the equivalent”? Equivalent to what?

Line 667: “, thus” should be “; thus,”

Lines 687-689: “make the strong assumption that imagery ability in aphantasia can be upregulated by attention or similar factors”. Why? Maybe such participants used a non-imagery strategy. Just because an imagery strategy is one possibly strategy does not mean that individuals necessarily use that strategy.

Line 690: “was less susceptible” seem a bit strong. I would be more cautious and say “was potentially less susceptible”. Also, after “characteristics, the sentence should be continued stating why such a claim is being made (i.e., “… characteristics’, because…).

Line 723: “Until this is achieved… highly heterogenous”. Well, participants could still be highly heterogeneous even after a uniform diagnostic criterion is developed.

*** end of review

7. PLOS authors have the option to publish the peer review history of their article (what does this mean?). If published, this will include your full peer review and any attached files.

Reviewer #1: No

Reviewer #2: **Yes: **Timothy L. Hubbard

---

## [Author Response · Author response to Decision Letter 1]

27 Nov 2023

Please find a revised version of our manuscript. We thank the reviewers for their reviews of our manuscript. We have highlighted the changes that have been made in the manuscript and provide a response to each comment below subsequent to this round of review. 

Response to reviewer comments

Reviewer #1: The authors fully addressed most of my previous comments.

The only point I think could due with some additional discussion is the correlations between the questionnaires and the auditory pitch task. If I understood this correctly, these two measurements are positively correlated in the control group – suggesting that subjective reports of vividness correlate with objective performance in this task – but there is no difference between the aphantasic and control groups, which are themselves defined based on the subjective reports (low VVIQ versus high VVIQ). 

The authors suggest that this might be because ‘aphantasic participants were more greatly motivated to perform well on this task given their self-reported difficulties, which acted to compensate for their imagery deficits.’ But this seems a little far-fetched to me. Are other explanations possible? For example, perhaps aphantasic participants use different strategies to perform the same task (e.g. as suggested by Koegh, Wicken & Pearson, 2021 for working memory). Could the authors speculate more on this?

We agree that this is unlikely, but we were asked to speculate on whether ‘demand characteristics’ could have influenced our findings in the previous round of review. Given that we are reporting null results, this was one way that “demand characteristics” could have influenced the pattern of results, under the expectation that aphantasics should ordinarily perform worse than control participants. We have changed the emphasis and incorporated your suggestion, it now reads:

“Given that this task explicitly asked aphantasic participants to generate auditory imagery, it may have been subject to ‘demand characteristics’. However, one might assume that such a bias would lead to reduced rather than increased performance in the aphantasic group. Given that performance was equivalent between the groups, it may be that participants with aphantasia drew upon alternative non-imagery-based strategies to complete the tasks”

Reviewer #2: *** beginning of review

General issues:

Relationship of Aphantasia to Perception. The authors address the possibility of aphantasia across different modalities in imagery. Even so, I found myself wondering (e.g., while reading lines 70-83) about the relationship between aphantasia in a given modality and perceptual functioning within that same modality. Given that there is a correlation between patterns of brain activation during perception in a given modality and imagery in that same modality, one suspects aphantasia might be related to deficits in perception. This would also seem related to the question of whether imagery draws on a single unitary mechanism or multiple cognitive mechanisms (cf. line 83). Although the relationship between imagery and perception is briefly addressed on lines 143-149, perhaps this might be moved to earlier in the introduction and expanded. Indeed, if auditory aphantasia is not linked to any auditory perceptual deficits, then that would have significant implications for understanding such aphantasia, as well as understanding of imagery more generally.

We thank the reviewer for this suggestion. However, we feel that this point is already adequately expressed in lines 143-149 (as pointed out by the reviewer). We would prefer not to change the structure of the introduction of our manuscript to accommodate this request. We feel that this point is already clearly made and as such do not want to change the existing structure and organisation of the arguments in the introduction section. We leave it to the action editor to decide whether this change is required. 

Specifying the Modality. There are multiple instances in which in the authors refer to “mental images”. Sometimes it is not clear if they are referring to both visual images, auditory images, or both (unimodal) visual and auditory images (e.g., lines 419, 421, 448, etc.). Relatedly, visual imagery and auditory imagery are treated as separate types of unimodal images, but would the same patterns of aphantasia occur if a single multimodal image were generated? Also, I would suggest that rather than the general “mental imagery” the authors always use the more specific “visual mental imagery” or “auditory mental imagery” whenever possible. There are also a few instances in which the authors refer to “aphantasia participants”, but it isn’t clear whether the aphantasia being referred to is visual or auditory (e.g., line 533). 

Where appropriate, we have made sure that we have specified if the imagery in reference is visual or auditory. When we discuss imagery ‘in general’ we refer to imagery without mention of the specific modality of the imagery.

“Control Group” vs. “Typical Imagery Group”. The authors seem to alternative between referring to their control group as the “control group” (e.g., lines 388, 390, 413, 458, etc.) and referring to their control group as the “typical imagery group” (e.g., lines 448, 453, 456, 476, etc.). Using multiple terms for the same construct or group makes more work for the reader, and so the authors should just pick a single term and use that term more consistently.

We now consistently refer to the ‘typical imagery control group’ to reduce the use of multiple terms.

Specific Issues:

Lines 35-36: Line 35 states “it is possible to generate many different kinds of internal representation”, which implies generation of non-imagery forms of representation as well as generation of imagery, but all the examples listed on line 36 seem to involve imagery.

We have changed this to refer to imagery experiences only. 

Lines 61-62: “are well documented” might be a bit strong. Perhaps “are relatively well documented” might be better.

This has been changed.

Line 78: Perhaps “in those domains” should be inserted after “require imagery”.

This has been added.

Line 105: What are “frank differences” and how do they differ from just “differences”? If there is a difference, it should be explained, and if there is no difference, perhaps “frank” should be deleted. Relatedly, what is a “frank deficit” (on line 168), and how does that differ from just a “deficit”? If there is a difference, it should be explained, and if there is no difference, perhaps “frank” should be deleted.

The word ‘frank’ has been removed from both of these sentences.

Lines 106-110: The statement “it is not clear… one explanation could be…” seems disorganized. More specifically, it seems strange to me that the authors start the second sentence with “one explanation…” given that they just spend lines 90-105 discussing other possible explanations for why differences were not more apparent. (e.g., imagery is epiphenomenal, qualitatively different or weaker imagery, etc.). Rather than making line 106 sound like a shift to a different topic (an explanation rather than a description), it might be better to make it clearer that additional explanations will be given.

We have added to the end of the first sentence that there are several possible explanations, and changed the beginning of the following sentence (that begun “One explanation”) to “For example”. For clarity, this reads:

“It is not clear why differences in performance on imagery tasks are not more apparent in aphantasic groups, however, there are several possible explanations. For example, One explanation could be that people with aphantasia use compensatory…”

Lines 130-134: A greater number of beats in an auditory image and a longer distance in a visual image are stimulus-specific properties of the images, and it isn’t entirely clear why a common mechanism would respond to these different types of stimulus-specific properties, and so perhaps some speculation on this might be offered. 

We have added an additional example of mental rotation to support this argument and we have also added an additional sentence that speculates at a common mechanism. This reads:

For example, it takes longer to scan through an imagined melody that contains a longer series of beats (Halpern, 1988; Zatorre & Halpern, 2005) in a manner analogous to the increased time that it takes to visually scan imagined objects that are further apart in distance (Kosslyn, 1981; Pearson et al., 2015) or rotate objects that require larger angular rotations (Shepard & Metzler, 1971). This suggests that both auditory and visual imagery may be underpinned by common spatial mechanisms that allow for auditory images to be extended in time akin to the way visual images are extended in space.

Also, for consistency “ones” in line 130 should be replaced with “images”.

This has been changed.

Line 140: For clarity, I would add “the amount of” before “grey”

This has been added.

Lines 173-181: Was the VVIQ score used in recruiting participants, or was it only obtained after recruitment? The text implies the former, but I suspect the latter actually occurred. Regardless, this should be clarified. Also, did the forums from which the authors recruited participants distinguish between visual or auditory aphantasia? More specifically, did the participants actually self-identify as aphantastic based on their visual imagery, or might some self-identify as aphantastic based on their auditory imagery? Also, would not have recruiting on the basis of auditory aphantasia (rather than recruiting based on visual aphantasia) have also allowed examination of the relationship between visual aphantasia and auditory aphantasia? Other than possibly having more total respondents if they advertised for visual aphantasia, why didn’t the authors recruit from the population that they were really interested in, which was auditory aphantastics?

The VVIQ was obtained following recruitment, we have referred to each groups’ VVIQ score following the descriptions of how the two groups were recruited. 

There were theoretical and practical reasons for conducting recruitment in the way that we did.

This study was run from September 2019- March 2020, during this time, aphantasia was defined as an inability to generate visual imagery (Dawes et al., 2020; Zeman et al., 2015, Zeman et al., 2020), and more recent definitions have extended this definition to include the absence of other sensory modalities (Monzel et al., 2022), with terms such as “auditory aphantasia” or “visual aphantasia” commonly used to describe the absence of imagery within specific sensory modalities. We have added to the text the time frame of when this data was collected, as this is important in the context of aphantasia as the definition of aphantasia has altered within the last year. Our approach was motivated by the desire to understand the extent of the co-occurrence of auditory and visual imagery deficits in people that call themselves “aphantasic”. 

Aphantasia is a niche population (experienced by less than 5% of the population). It is difficult to recruit for studies of people with aphantasia for this reason. Thanks to a growing recognition of the condition we are lucky to be able to recruit via various aphantasia interest groups. These groups do not currently exist for people with “auditory imagery deficits”. 

We have added the following to acknowledge the shift in aphantasia definitions and contextualise our study in relation to these studies prior to this definition shift: 

“This study was undertaken between September 2019- March 2020, during which, aphantasia was defined specifically as the absence of visual imagery (Zeman et al., 2015; Zeman et al., 2020) rather than an absence of sensory imagery (Monzel et al., 2022). We recruited participants on this basis to investigate the association of auditory and visual imagery deficits in individuals that would typically identify as being aphantasic (on the basis of their reduced visual imagery). This approach was taken to investigate the auditory imagery of individuals that would typically be identified as ‘aphantasic’ in previous research studies in this area that were collected during this time.”

We also draw attention to the fact that we explicitly discuss whether this recruitment strategy may have influenced the findings in our discussion.

Line 176, 217, 298: I think using the Oxford comma usually improves clarity, and so I would insert a comma after “Twitter” on line 176, after “preferences” on line 217, and after “emotion” on line 298. Of course, if I missed any lists of three or more items that didn’t already include an Oxford comma, then analogous comments would apply to those lists.

We have added these commas and added them where appropriate. 

Lines 209-212: I realize the data for the control subscale will eventually be discarded, but this subscale should still be fully described. An example question from the subscale should be given and the types of transformations described. Also, anchor terms for both the vividness subscale and the control subscale should be provided.

We have amended this description of the BAIS-C subscale and provided the anchor descriptions for both scales. The BAIS description reads as follows:

“The BAIS (Halpern, 2015) is an auditory imagery questionnaire that comprises two scales: the vividness (BAIS-V) and control (BAIS-C) subscales. The vividness subscale requires participants to reflect on the vividness with which they can imagine 14 sounds described in a written scenario (e.g. “consider attending a choir rehearsal, the sound of an all children’s choir singing the first verse of a song”) on a scale of 1 ("no image present at all”) to 7 (“as vivid as actual sound”). These scenarios involve imagining different kinds of sounds that include voices and musical instruments. The control subscale comprises of 14 pairs of imagined sounds, which require participants to rate ‘the ease of change’ or transition on a scale of 1 (“no image present at all”) to 7 (“extremely easy to change the image”) between two imagined auditory scenarios (e.g. “Consider listening to a rain storm. The sound of gentle rain. The gentle rain turns into a violent thunderstorm”).”

Line 218: A space should be inserted after “behaviour”.

This space has been added.

Lines 228-232: The authors refer to the General Musical Sophistication Scale as one part of the Goldsmiths Musical Sophistication Index, but when they describe scoring of the Scale, the claim “these values are summed to provide an overall score for the participants” sounds as if they are describing the Index (the Gold-MSI) rather than a separate General Musical Sophistication Scale. This needs to be clarified.

We have clarified this to ensure we are referring to the General Musical Sophistication Scale, this sentence now reads: 

“Scores are entered into the Gold-MSI General Musical Sophistication scoring template to obtain a normalised value for each item, and these values are summed to provide an overall general musical sophistication score for each participant.”

Line 258: For consistency with the latter part of the sentence, “underlined” should be inserted after “second”.

We have added ‘underlined’ to this sentence. 

Line 297: A space should be inserted after “properties”.

This space has been added.

Line 304: “can” should be inserted before “draw” or “draw” should be changed the “draws”. The former would be preferable.

This has been amended with the former request (we have changed the ‘can’ to ‘may’ following your comment below).

Lines 304-312: But is imagery really needed for the voice task? The voice task is intended to be an analogue of facial recognition, but the authors already admit that “the role of imagery in facial recognition remains unclear” (lines 281-282). I’m not suggesting that imagery can’t be used in the voice task; it might be that at least some participants used imagery at least some of the time. What I am suggesting is that it isn’t clear that imagery is required for the voice task, either for the aphantasia group or for the control group. Perhaps voices might be characterized based on descriptions of some features rather than imagery per se.

We agree with you that we cannot be sure that imagery is required to perform the voice task. As we stated previously, we used this task because we believed that imagery could be beneficial to task performance and because it is analogous to facial recognition, a domain that has been the subject of continuing research in aphantasia (Zeman et al., 2010; Dance et al., 2023; Milton et al., 2021). We have tested for the association between the voice task and voluntary visual and auditory imagery and found no evidence of a relationship. We report this and acknowledge in the discussion that imagery might not be necessary to do this task. We have now also added an additional sentence:

“However, we did not find evidence that performance on this task was correlated with auditory imagery self-report and as such it may not have been an effective task for measuring auditory imagery ability. This may be because participants were able to identify and use acoustic features as verbal labels to differentiate between speakers without needing to generate an auditory image.”

Lines 337-338: Why were different CV syllables used for make speakers and for female speakers? In other words, why wasn’t the same set of two (or four) CV syllables used for both male speakers and female speakers? 

We wanted to introduce different sets of CV syllables to provide variety in speaker characteristics and phonetic content over which vocal characteristics were expressed. We did not have an apriori hypothesis that the gender of the speaker would influence ease of imagery and so there was no need to balance the phonetic content across the speakers. 

Line 342: Maybe I just missed it, but what is a Bamford-Kowal-Bench sentence? How does it differ from other types of sentences? I appreciate that the authors added a reference, but this should still be clarified in the text.

We have added an example of a Bamford-Kowal-Bench (BKB) sentence in the manuscript. BKB sentences are short 4-6 word sentences, such as “the clown had a funny face”, “they ate the lemon jelly”. BKB sentences are designed to assess speech recognition of children who are hard of hearing. For your interest, the full list of BKB sentences can be found here: https://www.ssc.education.ed.ac.uk/courses/deaf/aud2bkb.html. We have added detail explaining that they are short sentences with a simple syntax and vocabulary:

“These are short 4-6 word sentences (e.g. “they ate the lemon jelly”) that have a simple syntax and vocabulary (58)”

Line 357: I’m not clear what “presented one by one in a fixed order to reduce confusion” means. Wouldn’t the pairs be presented one-at-a-time anyway? How does presenting a fixed order reduce confusion (e.g., were not the male voices clearly distinguishable from the female voices)? Could this fixed order supply or allow additional cues that might make imagery more or less necessary or useful?

This is a confusion that has arisen from removing the supplementary materials. We have know re-introduced this sentence which was originally in the supplementary materials into the methods section:

" The data from a third pair of speakers was collected but participants struggled to categorise the morphed continua into distinct vocal identities and so these data were not analysed further.”

Line 361: “each participant was tested one-by-one” should either be “participants were tested one-by-one” or “each participant was tested individually”. “each” implies a single participant, and you can’t have a single participant tested “one-by-one”.

We have changed this to ‘Each participant was tested individually”.

Line 362: The comma after “BAIS” should be deleted (or if not, then I don’t understand the sentence).

This comma has been removed.

Line 374-375: I’d insert a comma after “met” and delete the comma after “conducted”.

This has been amended.

Line 378-381: I still don’t think the authors need to list all the BF levels here. Listing interpretations for all BF values is like mentioning that p values greater than .05 won’t be considered significant in the presentations of the Mann-Whitney U tests or ANOVAs. Giving the interpretation of the BF that was actually obtained is sufficient. I leave this up to the action editor.

Given that this is only takes up a few lines within the manuscript, we would prefer to keep it in and defer to the action editor. We argue that many readers are not familiar with Bayes-Factors and this additional information is useful and does not take up much space. .

Line 393: “data… was”. Unless the authors are referring to the android in Star Trek, “data” is a plural term, and so “data… were” is correct (the singular form is “datum”).

This has been changed. 

Lines 437-442: This was confusing. The authors state that the analysis was carried out using d prime values (line 438), but the results were reported in terms of accuracy (line 441). What was the dependent variable? D prime? Or accuracy?

We have made it clearer that we are referring to d-prime values rather than accuracy (e.g. % or proportion correct). We no longer refer to accuracy to make sure that this is fully clear to readers.

 Lines 470-471: “performance on” should be inserted before both instances of “the pitch task”, as the correlation with imagery does not involve the task per se, but rather involves performance on the task.

This has been amended.

Line 487: Box and Cox (1964) is still not in the References section.

Apologies for this formatting error, this now appears in the reference section.

Lines 499-501: Whether the effect of participant group was significant or not should probably be mentioned, as well.

We have added a sentence to the summary of this paragraph to make it clear that we are referring to imagery effects specifically.

Line 517: If the authors wish to conform to APA style, then abbreviations such as “e.g.,” and “i.e.,” are used only within parenthetical expression. Within the main text, these should be spelled out (e.g., “for example”, “that is”, etc.)

We have re-formatted in Vancouver style to conform to the journals default referencing style.

Line 525: Although the readers question of “95th what?” is answered by the end of the sentence, just seeing “above the 95th” does seem a little odd.

We have edited this sentence so that the 95th is in reference to the winsorized values, and this reads: 

“…recoding all winsorized values above the 95th to the value attained at the 95% percentile”

Line 543: How was the combining of VVIQ and BAIS-V into a composite score done? Were the individual scores added, averaged, or was some other method of combination used?

The composite scores were created by calculating the mean of z-scored values for each task and then averaging the z scores. The explanation is explained once earlier in the manuscript. There is a second reference to this process later on in the results (this time for the voice task) which we think this is referring to. We have now added the following text so that it is clear, when we reference it this second time:

“There was no evidence of a relationship between auditory or visual imagery self-report and performance on the voice task alone or when the imagery measures were combined in a composite score by z-scoring and then averaging them”

Line 546: “BAISV” has been hyphenated previously in the manuscript (e.g., lines 393, 406, 411, 423-426, 541, etc.), and so should be hyphenated here for consistency.

This has been amended.

Line 573: Well, the tasks might not actually require auditory imagery if the objective performance on those tasks by people who claim to not experience auditory imagery does not differ from the objective performance by people who do claim to experience auditory imagery. Just because imagery might offer one strategy to complete a task does not mean that participants necessarily use that strategy (e.g., see Zatorre & Halpern’s, 2005, and Hubbard’s, 2010, argument why simply observing a given pattern of responses when participants are instructed to use imagery does not establish that those participants did in fact use imagery; see also Hubbard’s, 2018, discussion of representational ambiguity). Claiming that the tasks required auditory imagery is probably too strong, and it would be much more prudent to weaken this.

We have softened this sentence, and this reads: 

“The current study compared the performance of individuals with aphantasia to participants with typical imagery on behavioural tasks that we thought would require auditory imagery.”

Lines 610-611: Perhaps this can be elaborated?

We have added this text:

“However, we investigated this empirically by testing to see if there was evidence that the distribution of auditory imagery scores in the aphantasic participants differed from a unimodal distribution (e.g., whether it reflected a sub-group with good auditory imagery and another with poor auditory imagery) and did not find evidence to support this.” 

Lines 631-632: But the differences are evident. Perhaps the authors intended to say that the reasons for the differences, rather than the differences per se, are not evident?

We have changed the wording in this sentence to the following:

“There may be several reasons as to why the differences are not evident in the current study.”

Line 639: There is an extra opening parenthesis on this line.

This has been removed. 

Lines 650-651: Plack et al. and Yuskaitis et al. are cited in the main text but are not listed in the References section.

Apologies for the formatting error, these now appear in the reference list.

Lines 658-660: “suggest” should be “suggests”. Also, an example of such evidence (even if parenthetical) would be helpful to readers.

We have amended the word ‘suggest’ to ‘suggests’ and added the following sentence as an example of this evidence:

“For instance, individuals with amusia have difficulties discriminating changes and pitch and perform worse on spatial tasks such as mental rotation compared to neurotypical individuals (Douglas & Bilkey, 2007; Tao et al., 2015).”

Line 663: What is “the equivalent”? Equivalent to what?

We have clarified this as a ‘spatial equivalent’.

Line 667: “, thus” should be “; thus,”

This has been changed.

Lines 687-689: “make the strong assumption that imagery ability in aphantasia can be upregulated by attention or similar factors”. Why? Maybe such participants used a non-imagery strategy. Just because an imagery strategy is one possibly strategy does not mean that individuals necessarily use that strategy.

We agree, we think primarily aphantasic participants used non-imagery strategies in the tasks, and we have adjusted this sentence accordingly. This now reads as follows: 

“Given that this task explicitly asked aphantasic participants to generate auditory imagery, it may have been subject to ‘demand characteristics’. However, one might assume that such a bias would lead to reduced rather than increased performance in the aphantasic group. Given that performance was equivalent between the groups, it may that participants with aphantasia drew upon alternative non-imagery-based strategies to complete the tasks”

Line 690: “was less susceptible” seem a bit strong. I would be more cautious and say “was potentially less susceptible”. Also, after “characteristics, the sentence should be continued stating why such a claim is being made (i.e., “… characteristics’, because…).

We have included the word potentially and reminded the reader as to why our task was less susceptible to demand characteristics, this reads:

“Our voice task was potentially less susceptible to ‘demand characteristics’ because we did not explicitly instruct participants to engage in auditory imagery during the task.”

Line 723: “Until this is achieved… highly heterogenous”. Well, participants could still be highly heterogeneous even after a uniform diagnostic criterion is developed.]

Yes, we agree, we have removed this sentence from the manuscript.

---

## [Decision Letter · Decision Letter 2]

13 Dec 2023

PONE-D-23-05751R2No clear evidence of a difference between individuals who self-report an absence of auditory imagery and typical imagers on auditory imagery tasksPLOS ONE

Dear Dr. Evans,

Thank you for submitting your manuscript to PLOS ONE. After careful consideration, we feel that it has merit but does not fully meet PLOS ONE’s publication criteria as it currently stands. Therefore, we invite you to submit a revised version of the manuscript that addresses the points raised during the review process.

We look forward to receiving your revised manuscript.

Kind regards,

Jie Wang, Ph.D.

Academic Editor

PLOS ONE

Journal Requirements:

Additional Editor Comments :

The authors should address the issues raised by Reviewer 2.

Reviewers' comments:

Reviewer's Responses to Questions

**Comments to the Author**

1. If the authors have adequately addressed your comments raised in a previous round of review and you feel that this manuscript is now acceptable for publication, you may indicate that here to bypass the “Comments to the Author” section, enter your conflict of interest statement in the “Confidential to Editor” section, and submit your "Accept" recommendation.

Reviewer #1: All comments have been addressed

Reviewer #2: (No Response)

2. Is the manuscript technically sound, and do the data support the conclusions?

Reviewer #1: Yes

Reviewer #2: Yes

3. Has the statistical analysis been performed appropriately and rigorously? 

Reviewer #1: Yes

Reviewer #2: Yes

4. Have the authors made all data underlying the findings in their manuscript fully available?

Reviewer #1: (No Response)

Reviewer #2: Yes

5. Is the manuscript presented in an intelligible fashion and written in standard English?

Reviewer #1: Yes

Reviewer #2: Yes

6. Review Comments to the Author

Reviewer #1: All my comments have been addressed.

Reviewer #2: General Issues:

Headings. I have a concern with the heading levels. In a single experiment paper, “Methods”, “Results”, and “Discussion” are the highest level headings. Subheadings such as “Participants”, “Stimuli” in the Methods section and subheadings for different tasks or analyses in the Results section should be at a lower heading level. However, all the headings in the main text appear to be at the same level.

Gold-MSI. Some parts of the description of the Gold-MSI (lines 219-232) are still confusing to me. Based on the sentence structure, I would normally interpret “self-reported singing ability and sophisticated emotional engagement” as two items, but the lack of a comma after “ability” could be read as suggesting that these might be different aspects of a single item. When I count the items on lines 220-224, I got 5 dimensions only if I consider self-reported singing ability and sophisticated emotional engagement as separate items (but readers shouldn’t have to count to resolve an ambiguity). Furthermore, I’m not entirely clear how “sophisticated emotional engagement" differs from general musical sophistication (although I admit this might just reflect a poor choice of names for the dimensions and not due to the current authors). Lines 226-227 implies there are 18 questions, but it isn’t clear whether this is the total number of questions in the survey or the number of questions on that dimension (and if the former, how many questions are there on each of the 5 dimensions?).

Specific Issues:

Line 37: “beyond” has connotations regarding spatial localization, and so “other than” or “in addition to” might be better.,

Lines 58-60: I would consider reversing the order of these first two sentences, as it makes more intuitive sense (to me, at least) to first describe a general phenomenon (imagery) before describing a limitation to that phenomenon (aphantasia) than to first describe a limitation to a general phenomenon (aphantasia) and then describe the general phenomenon (imagery).

Line 61: A space should be inserted after “documented”.

Line 83: As “shown” was used on line 81 prior to the list, “show” should be deleted on line 83.

Line 85: “imagery behavior” is potentially confusing, as imagery is generally considered internal and not observable by others, whereas behavior is generally considered external and observable by others. I wound delete “behavior”.

Line 95: Although grouping all the references for a series of claims at the end of a sentence is permissible, I (and probably other readers) find it more useful if a citation is given after each claim (as the authors did on lines 82-84), and I would suggest they place the relevant citation after each claim on lines 92-95 (and elsewhere in manuscript as relevant) unless of course each of the references supports each of the claims (in which case placing all the references at the end of the sentence is better).

Line 102: The comma before “however” should probably be a semi-colon.

Line 108: I think this might be clearer if “else” is inserted before “draw”.

Line 136: As I noted in my previous review, several style guides suggest that it is not good style to begin a paragraph with “However”. I would delete “However,”

Line 138: Commas should be inserted before and after “respectively”.

Line 160: I would consider inserting “then” before “they would”.

Line 171: I would suggest changing “the” to “that”.

Lines 175-176: “in previous research studies in this area that were collected during this time” is confusing. It isn’t clear whether “this time” refers to when the previous studies were done or to the time of the current study. Moreover, how could previous research be carried out during the current time?

Line 179: I’m not sure this comma is needed.

Lines 180-182: Even if the cutoff was used and described in a previous study, readers shouldn’t have to go search for a previous document for an important piece of methodological information. The cutoff should be stated in the current manuscript (and of course, it is fine to state that the same cutoff was used in previous studies).

Line 248: I would add “each” after “20 trials”

Lines 309, 680, 685, etc.: It isn’t necessary to use quote marks around “demand characteristics”, as the idea of demand characteristics (and the need to control for demand characteristics) has a long history in psychological research, and especially in the study of imagery (see reviews of auditory imagery such as Hubbard, 2010, Psych Bull; 2018, AP&C).

Lines 312-356: This paragraph is too long and needs to be broken into 2-3 smaller paragraphs (which might necessitate some reorganization).

Lines 326-328: How does the “number of continua” in line 326 relate to the “4 continua” in line 328? If they are the same, then perhaps these two sentences can be consolidated.

Line 330: Perhaps “between group” should be “between-group”.

Line 335: A space needs to be inserted before “Prior”.

Line 348: I might add a comma after “crucially”.

Lines 360-362: As I noted in my previous review, several style guides recommend use of the Oxford comma (i.e., a comma after the penultimate item in a list) to maximize clarity. Accordingly, there should be a comma inserted after “Gold-MSI” and after “pitch task”. A similar lack of an Oxford comma contributed to the confusion noted in General Issue B).

Line 374: “that” should be deleted.

Lines 375-378: I still don’t think it’s necessary to report how any possible BF would be interpreted, but the authors clearly disagree. Nonetheless, this is a minor issue, and I won’t insist on it, but will point out that other readers might find it odd.

Lines 390-395: I presume the change from full to left justification for this paragraph is a typo, but it should be corrected, nonetheless.

Lines 408-409: Shouldn’t “vividness of” be inserted after “between” and also before “auditory imagery”? After all, the two scales being used (VVIQ, BAIS-V) focus on vividness and not on other characteristics of the image (and previously noted, Lacey and Lawson [2013] have criticized use of “vividness” as a general evaluation of imagery, as it does not consider several other aspects of imagery [generation, manipulation, control, etc.]).

Lines 410-412: It seems rather odd that something that shouldn’t exist (i.e., imagery in aphantasia) should be correlated with anything.

Lines 553-570: This paragraph is the final paragraph under the “Voice task” subheading in the Results section, but it doesn’t seem to really fit under that subheading. Perhaps a new subheading should be inserted before line 553 (e.g., perhaps something like “Further Comparison of Aphantasia and Control Groups”)? Alternatively, maybe this paragraph could be footnoted at (or added to) line 402 when the result of the initial dip test was reported?

Line 605: I’m not sure the comma after “neglect” is correct. Actually, I would change this to “…conditions (e.g., hemi-spatial neglect), can…”.

Line 609: I would delete “However,”.

Line 688: Participants completed the VVIQ and BAIS questionnaires just before they did the pitch task and the voice task. It seems possible (at least in principle) that the focus on imagery in those questionnaires might have suggested to participants that imagery was an important part of the experiment. While it is good that imagery was not mentioned in the instructions to the voice task, that in itself (in light participants filling out the VVIQ and the BAIS just before completing the pitch task and the voice task) is not sufficient to rule out demand characteristics. This should also be addressed.

References: There are some inconsistences across the References section (e.g., some article titles capitalize the first letter of all of the “important" words, whereas other article title capitalize only the first letter of the first word; some journal names are abbreviated, but other journal names are spelled out in full; etc.).

*** end of review

7. PLOS authors have the option to publish the peer review history of their article (what does this mean?). If published, this will include your full peer review and any attached files.

Reviewer #1: No

Reviewer #2: No

---

## [Author Response · Author response to Decision Letter 2]

13 Feb 2024

Reviewer #1: All my comments have been addressed.

Reviewer #2: General Issues:

Headings. I have a concern with the heading levels. In a single experiment paper, “Methods”, “Results”, and “Discussion” are the highest level headings. Subheadings such as “Participants”, “Stimuli” in the Methods section and subheadings for different tasks or analyses in the Results section should be at a lower heading level. However, all the headings in the main text appear to be at the same level.

We are sure that further formatting of the manuscript will be undertaken by the journal's production team. However, to ensure readability for the reviewer, we have adjusted the headings so that the subheadings are in a smaller font than the ‘main’ headings. 

Gold-MSI. Some parts of the description of the Gold-MSI (lines 219-232) are still confusing to me. Based on the sentence structure, I would normally interpret “self-reported singing ability and sophisticated emotional engagement” as two items, but the lack of a comma after “ability” could be read as suggesting that these might be different aspects of a single item. When I count the items on lines 220-224, I got 5 dimensions only if I consider self-reported singing ability and sophisticated emotional engagement as separate items (but readers shouldn’t have to count to resolve an ambiguity). 

To ensure utter clarity and to remove any confusion, we have edited this section and we have numbered each dimension to clearly illustrate to the reader, the 5 dimensions.

Furthermore, I’m not entirely clear how “sophisticated emotional engagement" differs from general musical sophistication (although I admit this might just reflect a poor choice of names for the dimensions and not due to the current authors).

These are the names of the original dimensions, with the “general musical sophistication” scale including items from all 5 of the dimensions – unfortunately we cannot comment further on the naming convention for these scales. 

Lines 226-227 implies there are 18 questions, but it isn’t clear whether this is the total number of questions in the survey or the number of questions on that dimension (and if the former, how many questions are there on each of the 5 dimensions?).

There are a total of 18 questions. This is now clarified in the manuscript on Page 10.

Specific Issues:

Line 37: “beyond” has connotations regarding spatial localization, and so “other than” or “in addition to” might be better.

This has been changed to ‘other than’ on Page 2.

Lines 58-60: I would consider reversing the order of these first two sentences, as it makes more intuitive sense (to me, at least) to first describe a general phenomenon (imagery) before describing a limitation to that phenomenon (aphantasia) than to first describe a limitation to a general phenomenon (aphantasia) and then describe the general phenomenon (imagery).

We have reversed the order of these sentences on Page 3.

Line 61: A space should be inserted after “documented”.

This has been added on Page 3.

Line 83: As “shown” was used on line 81 prior to the list, “show” should be deleted on line 83.

This has been deleted on Page 4.

Line 85: “imagery behavior” is potentially confusing, as imagery is generally considered internal and not observable by others, whereas behavior is generally considered external and observable by others. I wound delete “behavior”.

We have deleted the word ‘behaviour’ from this sentence on Page 4.

Line 95: Although grouping all the references for a series of claims at the end of a sentence is permissible, I (and probably other readers) find it more useful if a citation is given after each claim (as the authors did on lines 82-84), and I would suggest they place the relevant citation after each claim on lines 92-95 (and elsewhere in manuscript as relevant) unless of course each of the references supports each of the claims (in which case placing all the references at the end of the sentence is better).

To avoid repetition of citations after each claim, we would like to keep the citations as a group. Indeed, other readers (including the authors) prefer a less cluttered presentation by placing all references together at the end.

Line 102: The comma before “however” should probably be a semi-colon.

This has been changed on Page 4.

Line 108: I think this might be clearer if “else” is inserted before “draw”.

This has been added on Page 5.

Line 136: As I noted in my previous review, several style guides suggest that it is not good style to begin a paragraph with “However”. I would delete “However,”

We have replaced ‘However’ with ‘Nevertheless’ on Page 6. 

Line 138: Commas should be inserted before and after “respectively”.

This has been added on Page 6.

Line 160: I would consider inserting “then” before “they would”.

This has been added on Page 7.

Line 171: I would suggest changing “the” to “that”.

This has been changed on Page 7.

Lines 175-176: “in previous research studies in this area that were collected during this time” is confusing. It isn’t clear whether “this time” refers to when the previous studies were done or to the time of the current study. Moreover, how could previous research be carried out during the current time?

We have clarified that we are referring to the time frame of September 2019- March 2020 on Page 8.

Line 179: I’m not sure this comma is needed.

This comma has been removed on Page 8.

Lines 180-182: Even if the cutoff was used and described in a previous study, readers shouldn’t have to go search for a previous document for an important piece of methodological information. The cutoff should be stated in the current manuscript (and of course, it is fine to state that the same cutoff was used in previous studies).

We have included the cut-offs in the second sentence, and signposted this so that it is explicitly clear what these cut-offs are on Page 8.

Line 248: I would add “each” after “20 trials”

This has been added on Page 11.

Lines 309, 680, 685, etc.: It isn’t necessary to use quote marks around “demand characteristics”, as the idea of demand characteristics (and the need to control for demand characteristics) has a long history in psychological research, and especially in the study of imagery (see reviews of auditory imagery such as Hubbard, 2010, Psych Bull; 2018, AP&C).

These have been removed throughout the manuscript.

Lines 312-356: This paragraph is too long and needs to be broken into 2-3 smaller paragraphs (which might necessitate some reorganization).

This paragraph has been broken down to three paragraphs on Page 13 and Page 14.

Lines 326-328: How does the “number of continua” in line 326 relate to the “4 continua” in line 328? If they are the same, then perhaps these two sentences can be consolidated.

We have consolidated this into one sentence, which reads as follows on Page 14:

“We used 4 continua to ensure that multiple acoustic features would contribute to voice identification and provide a variety in speaker characteristics and phonetic content over which vocal characteristics were expressed.”

Line 330: Perhaps “between group” should be “between-group”.

This has been changed on Page 14.

Line 335: A space needs to be inserted before “Prior”.

This is now a new paragraph.

Line 348: I might add a comma after “crucially”.

This has been added on Page 15.

Lines 360-362: As I noted in my previous review, several style guides recommend use of the Oxford comma (i.e., a comma after the penultimate item in a list) to maximize clarity. Accordingly, there should be a comma inserted after “Gold-MSI” and after “pitch task”. A similar lack of an Oxford comma contributed to the confusion noted in General Issue B).

We have made this change although as the reviewer notes this is a style preference. 

Line 374: “that” should be deleted.

This has been deleted.

Lines 375-378: I still don’t think it’s necessary to report how any possible BF would be interpreted, but the authors clearly disagree. Nonetheless, this is a minor issue, and I won’t insist on it, but will point out that other readers might find it odd.

Thanks, but we do think this is a worthwhile and an important addition.

Lines 390-395: I presume the change from full to left justification for this paragraph is a typo, but it should be corrected, nonetheless.

This has been adjusted. 

Lines 408-409: Shouldn’t “vividness of” be inserted after “between” and also before “auditory imagery”? After all, the two scales being used (VVIQ, BAIS-V) focus on vividness and not on other characteristics of the image (and previously noted, Lacey and Lawson [2013] have criticized use of “vividness” as a general evaluation of imagery, as it does not consider several other aspects of imagery [generation, manipulation, control, etc.]).

“Vividness of” has been added twice to this sentence, which reads as follows on Page 17:

“We further investigated the association between self-reported vividness of visual imagery (measured by the VVIQ) and vividness of auditory imagery (measured by the BAIS-V).”

Lines 410-412: It seems rather odd that something that shouldn’t exist (i.e., imagery in aphantasia) should be correlated with anything.

Conceptually one could argue this, however, the aim of this analysis is to inform the reader that low vividness as represented by scores on one questionnaire for one sensory modality (i.e. vision) correlates with low vividness scores in the second sensory modality (i.e. audition).

Lines 553-570: This paragraph is the final paragraph under the “Voice task” subheading in the Results section, but it doesn’t seem to really fit under that subheading. Perhaps a new subheading should be inserted before line 553 (e.g., perhaps something like “Further Comparison of Aphantasia and Control Groups”)? Alternatively, maybe this paragraph could be footnoted at (or added to) line 402 when the result of the initial dip test was reported?

We have added the heading “Further Comparison of Aphantasia and Control Groups” to this section on Page 24.

Line 605: I’m not sure the comma after “neglect” is correct. Actually, I would change this to “…conditions (e.g., hemi-spatial neglect), can…”.

This has been edited on Page 26.

Line 609: I would delete “However,”.

This has been removed on Page 27.

Line 688: Participants completed the VVIQ and BAIS questionnaires just before they did the pitch task and the voice task. It seems possible (at least in principle) that the focus on imagery in those questionnaires might have suggested to participants that imagery was an important part of the experiment. While it is good that imagery was not mentioned in the instructions to the voice task, that in itself (in light participants filling out the VVIQ and the BAIS just before completing the pitch task and the voice task) is not sufficient to rule out demand characteristics. This should also be addressed.

We think this is unlikely, however, we have added the following sentence to the discussion on Page 29 to address this possibility:

“Further, all questionnaires were undertaken prior to undertaking the task, which may have also suggested to participants that imagery was necessary for task performance.”

References: There are some inconsistences across the References section (e.g., some article titles capitalize the first letter of all of the “important" words, whereas other article title capitalize only the first letter of the first word; some journal names are abbreviated, but other journal names are spelled out in full; etc.).

This is now corrected.

*** end of review

---

## [Decision Letter · Decision Letter 3]

26 Feb 2024

No clear evidence of a difference between individuals who self-report an absence of auditory imagery and typical imagers on auditory imagery tasks

PONE-D-23-05751R3

Dear Dr. Evans,

We’re pleased to inform you that your manuscript has been judged scientifically suitable for publication and will be formally accepted for publication once it meets all outstanding technical requirements.

Kind regards,

Jie Wang, Ph.D.

Academic Editor

PLOS ONE

Additional Editor Comments (optional):

Reviewers' comments:

Reviewer's Responses to Questions

**Comments to the Author**

1. If the authors have adequately addressed your comments raised in a previous round of review and you feel that this manuscript is now acceptable for publication, you may indicate that here to bypass the “Comments to the Author” section, enter your conflict of interest statement in the “Confidential to Editor” section, and submit your "Accept" recommendation.

Reviewer #1: All comments have been addressed

2. Is the manuscript technically sound, and do the data support the conclusions?

Reviewer #1: Yes

3. Has the statistical analysis been performed appropriately and rigorously? 

Reviewer #1: Yes

4. Have the authors made all data underlying the findings in their manuscript fully available?

Reviewer #1: (No Response)

5. Is the manuscript presented in an intelligible fashion and written in standard English?

Reviewer #1: Yes

6. Review Comments to the Author

Reviewer #1: (No Response)

7. PLOS authors have the option to publish the peer review history of their article (what does this mean?). If published, this will include your full peer review and any attached files.

Reviewer #1: No

---

## [Editor Report · Acceptance letter]

25 Mar 2024

PONE-D-23-05751R3 

PLOS ONE

Dear Dr. Evans, 

I'm pleased to inform you that your manuscript has been deemed suitable for publication in PLOS ONE. Congratulations! Your manuscript is now being handed over to our production team.

Kind regards, 

on behalf of

Dr. Jie Wang 

Academic Editor

PLOS ONE